# CARD: Classification and Regression Diffusion Models

**Xizewen Han**[*]   **Huangjie Zheng**[*]
Department of Statistics and Data Sciences
The University of Texas at Austin
Austin, TX 78712
{xizewen.han, huangjie.zheng}@utexas.edu

**Mingyuan Zhou**
McCombs School of Business
The University of Texas at Austin
Austin, TX 78712
mingyuan.zhou@mccombs.utexas.edu

## Abstract

Learning the distribution of a continuous or categorical response variable $y$ given its covariates $x$ is a fundamental problem in statistics and machine learning. Deep neural network-based supervised learning algorithms have made great progress in predicting the mean of $y$ given $x$, but they are often criticized for their ability to accurately capture the uncertainty of their predictions. In this paper, we introduce classification and regression diffusion (CARD) models, which combine a denoising diffusion-based conditional generative model and a pre-trained conditional mean estimator, to accurately predict the distribution of $y$ given $x$. We demonstrate the outstanding ability of CARD in conditional distribution prediction with both toy examples and real-world datasets, the experimental results on which show that CARD in general outperforms state-of-the-art methods, including Bayesian neural network-based ones that are designed for uncertainty estimation, especially when the conditional distribution of $y$ given $x$ is multi-modal. In addition, we utilize the stochastic nature of the generative model outputs to obtain a finer granularity in model confidence assessment at the instance level for classification tasks. Our implementation is publicly available at https://github.com/XzwHan/CARD.

## 1   Introduction

A fundamental problem in statistics and machine learning is to predict the response variable $y$ given a set of covariates $x$. Generally speaking, $y$ is a continuous variable for regression analysis and a categorical variable for classification. Denote $f(x) \in \mathbb{R}^C$ as a deterministic function that transforms $x$ into a $C$ dimensional output. Denote $f_c(x)$ as the $c$-th dimension of $f(x)$. Existing methods typically assume an additive noise model: for regression analysis with $y \in \mathbb{R}^C$, one often assumes $y = f(x) + \epsilon$, $\epsilon \sim \mathcal{N}(0, \Sigma)$, while for classification with $y \in \{1, \ldots, C\}$, one often assumes $y = \arg\max\left(f_1(x) + \epsilon_1, \ldots, f_C(x) + \epsilon_C\right)$, where $\epsilon_c \overset{iid}{\sim} \text{EV}_1(0, 1)$, a standard type-1 extreme value distribution. Thus we have the expected value of $y$ given $x$ as $\mathbb{E}[y \,|\, x] = f(x)$ in regression and $P(y = c \,|\, x) = \mathbb{E}[y = c \,|\, x] = \text{softmax}_c\left(f(x)\right) = \frac{\exp(f_c(x))}{\sum_{c'=1}^C \exp(f_{c'}(x))}$ in classification.

These additive-noise models are primarily focusing on accurately estimating the conditional mean $\mathbb{E}[y \,|\, x]$, while paying less attention to whether the noise distribution can accurately capture the uncertainty of $y$ given $x$. For this reason, they may not work well if the distribution of $y$ given $x$ clearly deviates from the additive-noise assumption. For example, if $p(y \,|\, x)$ is multi-modal, which commonly happens when there are missing categorical covariates in $x$, then $\mathbb{E}[y \,|\, x]$ may not be close to any possible true values of $y$ given that specific $x$. More specifically, consider a person whose weight, height, blood pressure, and age are known but gender is unknown, then the testosterone or estrogen level of this person is likely to follow a bi-modal distribution and the chance

---

[*]Equal contribution.

of developing breast cancer is also likely to follow a bi-modal distribution. Therefore, these widely used additive-noise models, which use a deterministic function $f(\boldsymbol{x})$ to characterize the conditional mean of $\boldsymbol{y}$, are inherently restrictive in their ability for uncertainty estimation.

In this paper, our goal is to accurately recover the full distribution of $\boldsymbol{y}$ conditioning on $\boldsymbol{x}$ given a set of $N$ training data points, denoted as $\mathcal{D} = \{(\boldsymbol{x}_i, \boldsymbol{y}_i)\}_{1,N}$. To realize this goal, we consider the diffusion-based (a.k.a. score-based) generative models (Sohl-Dickstein et al., 2015; Song and Ermon, 2019; Ho et al., 2020; Song and Ermon, 2020; Song et al., 2021c) and inject covariate-dependence into both the forward and reverse diffusion chains. Our method can model the conditional distribution of both continuous and categorical $\boldsymbol{y}$ variables, and the algorithms developed under this method will be collectively referred to as **C**lassification **A**nd **R**egression **D**iffusion (CARD) models.

Diffusion-based generative models have received significant recent attention due to not only their ability to generate high-dimensional data, such as high-resolution photo-realistic images, but also their training stability. They can be understood from the perspective of score matching (Hyvärinen and Dayan, 2005; Vincent, 2011; Kadkhodaie and Simoncelli, 2021) and Langevin dynamics (Neal, 2011; Welling and Teh, 2011), as pioneered by Song and Ermon (2019). They can also be understood from the perspective of diffusion probabilistic models (Sohl-Dickstein et al., 2015; Ho et al., 2020), which first define a forward diffusion to transform the data into noise and then a reverse diffusion to regenerate the data from noise.

These previous methods mainly focus on unconditional generative modeling. While there exist guided-diffusion models (Song and Ermon, 2019; Song et al., 2021c; Dhariwal and Nichol, 2021; Nichol et al., 2022; Ramesh et al., 2022) that target on generating high-resolution photo-realistic images that match the semantic meanings or content of the label, text, or corrupted-images, we focus on studying diffusion-based conditional generative modeling at a more fundamental level. In particular, our goal is to thoroughly investigate whether CARD can help accurately recover $p(\boldsymbol{y} \,|\, \boldsymbol{x}, \mathcal{D})$, the predictive distribution of $\boldsymbol{y}$ given $\boldsymbol{x}$ after observing data $\mathcal{D} = \{(\boldsymbol{x}_i, \boldsymbol{y}_i)\}_{i=1,N}$. In other words, our focus is on regression analysis of continuous or categorical response variables given their corresponding covariates.

We summarize our main contributions as follows: 1) We show CARD, which injects covariate-dependence and a pre-trained conditional mean estimator into both the forward and reverse diffusion chains to construct a denoising diffusion probabilistic model, provides an accurate estimation of $p(\boldsymbol{y} \,|\, \boldsymbol{x}, \mathcal{D})$. 2) We provide a new metric to better evaluate how well a regression model captures the full distribution $p(\boldsymbol{y} \,|\, \boldsymbol{x}, \mathcal{D})$. 3) Experiments on standard benchmarks for regression analysis show that CARD achieves state-of-the-art results, using both existing metrics and the new one. 4) For classification tasks, we push the assessment of model confidence in its predictions towards the level of individual instances, a finer granularity than the previous methods.

## 2 Methods and Algorithms for CARD

Given the ground-truth response variable $\boldsymbol{y}_0$ and its covariates $\boldsymbol{x}$, and assuming a sequence of intermediate predictions $\boldsymbol{y}_{1:T}$ made by the diffusion model, the goal of supervised learning is to learn a model such that the log-likelihood is maximized by optimizing the following ELBO:

$$\log p_\theta(\boldsymbol{y}_0 \,|\, \boldsymbol{x}) = \log \int p_\theta(\boldsymbol{y}_{0:T} \,|\, \boldsymbol{x}) d\boldsymbol{y}_{1:T} \geq \mathbb{E}_{q(\boldsymbol{y}_{1:T} \,|\, \boldsymbol{y}_0, \boldsymbol{x})} \left[ \log \frac{p_\theta(\boldsymbol{y}_{0:T} \,|\, \boldsymbol{x})}{q(\boldsymbol{y}_{1:T} \,|\, \boldsymbol{y}_0, \boldsymbol{x})} \right], \quad (1)$$

where $q(\boldsymbol{y}_{1:T} \,|\, \boldsymbol{y}_0, \boldsymbol{x})$ is called the forward process or diffusion process in the concept of diffusion models (Sohl-Dickstein et al., 2015; Ho et al., 2020). Denote $D_{\mathrm{KL}}(q \,\|\, p)$ as the Kullback–Leibler (KL) divergence from distribution $p$ to distribution $q$. The above objective can be rewritten as

$$\mathcal{L}_{\mathrm{ELBO}}(\boldsymbol{y}_0, \boldsymbol{x}) \coloneqq \mathcal{L}_0(\boldsymbol{y}_0, \boldsymbol{x}) + \sum_{t=2}^{T} \mathcal{L}_{t-1}(\boldsymbol{y}_0, \boldsymbol{x}) + \mathcal{L}_T(\boldsymbol{y}_0, \boldsymbol{x}), \quad (2)$$

$$\mathcal{L}_0(\boldsymbol{y}_0, \boldsymbol{x}) \coloneqq \mathbb{E}_q \left[ -\log p_\theta(\boldsymbol{y}_0 \,|\, \boldsymbol{y}_1, \boldsymbol{x}) \right], \quad (3)$$

$$\mathcal{L}_{t-1}(\boldsymbol{y}_0, \boldsymbol{x}) \coloneqq \mathbb{E}_q \left[ D_{\mathrm{KL}}\big(q(\boldsymbol{y}_{t-1} \,|\, \boldsymbol{y}_t, \boldsymbol{y}_0, \boldsymbol{x}) \,\big\|\, p_\theta(\boldsymbol{y}_{t-1} \,|\, \boldsymbol{y}_t, \boldsymbol{x})\big) \right], \quad (4)$$

$$\mathcal{L}_T(\boldsymbol{y}_0, \boldsymbol{x}) \coloneqq \mathbb{E}_q \left[ D_{\mathrm{KL}}\big(q(\boldsymbol{y}_T \,|\, \boldsymbol{y}_0, \boldsymbol{x}) \,\big\|\, p(\boldsymbol{y}_T \,|\, \boldsymbol{x})\big) \right]. \quad (5)$$

Here we follow the convention to assume $\mathcal{L}_T$ does not depend on any parameter and it will be close to zero by carefully diffusing the observed response variable $\boldsymbol{y}_0$ towards a pre-assumed distribution

$p(\boldsymbol{y}_T \mid \boldsymbol{x})$. The remaining terms will make the model $p_\theta(\boldsymbol{y}_{t-1} \mid \boldsymbol{y}_t, \boldsymbol{x})$ approximate the corresponding tractable ground-truth denoising transition step $q(\boldsymbol{y}_{t-1} \mid \boldsymbol{y}_t, \boldsymbol{y}_0, \boldsymbol{x})$ for all timesteps. Different from the vanilla diffusion models, we assume the endpoint of our diffusion process to be

$$p(\boldsymbol{y}_T \mid \boldsymbol{x}) = \mathcal{N}(f_\phi(\boldsymbol{x}), \boldsymbol{I}), \tag{6}$$

where $f_\phi(\boldsymbol{x})$ is the prior knowledge of the relation between $\boldsymbol{x}$ and $\boldsymbol{y}_0$, *e.g.*, a network pre-trained with $\mathcal{D}$ to approximate $\mathbb{E}[\boldsymbol{y} \mid \boldsymbol{x}]$, or $\boldsymbol{0}$ if we assume the relation is unknown. With a diffusion schedule $\{\beta_t\}_{t=1:T} \in (0, 1)^T$, we specify the forward process conditional distributions in a similar fashion as Pandey et al. (2022), but for all timesteps including $t = 1$:

$$q\big(\boldsymbol{y}_t \mid \boldsymbol{y}_{t-1}, f_\phi(\boldsymbol{x})\big) = \mathcal{N}\big(\boldsymbol{y}_t; \sqrt{1 - \beta_t}\boldsymbol{y}_{t-1} + (1 - \sqrt{1 - \beta_t})f_\phi(\boldsymbol{x}), \beta_t \boldsymbol{I}\big), \tag{7}$$

which admits a closed-form sampling distribution with an arbitrary timestep $t$:

$$q\big(\boldsymbol{y}_t \mid \boldsymbol{y}_0, f_\phi(\boldsymbol{x})\big) = \mathcal{N}\big(\boldsymbol{y}_t; \sqrt{\bar{\alpha}_t}\boldsymbol{y}_0 + (1 - \sqrt{\bar{\alpha}_t})f_\phi(\boldsymbol{x}), (1 - \bar{\alpha}_t)\boldsymbol{I}\big), \tag{8}$$

where $\alpha_t := 1 - \beta_t$ and $\bar{\alpha}_t := \prod_t \alpha_t$. Note that the mean term in Eq. (7) can be viewed as an interpolation between true data $\boldsymbol{y}_0$ and the predicted conditional expectation $f_\phi(\boldsymbol{x})$, which gradually changes from the former to the latter throughout the forward process.

Such formulation corresponds to a tractable forward process posterior:

$$q(\boldsymbol{y}_{t-1} \mid \boldsymbol{y}_t, \boldsymbol{y}_0, \boldsymbol{x}) = q\big(\boldsymbol{y}_{t-1} \mid \boldsymbol{y}_t, \boldsymbol{y}_0, f_\phi(\boldsymbol{x})\big) = \mathcal{N}\Big(\boldsymbol{y}_{t-1}; \tilde{\boldsymbol{\mu}}\big(\boldsymbol{y}_t, \boldsymbol{y}_0, f_\phi(\boldsymbol{x})\big), \tilde{\beta}_t \boldsymbol{I}\Big), \tag{9}$$

where

$$\tilde{\boldsymbol{\mu}} := \underbrace{\frac{\beta_t \sqrt{\bar{\alpha}_{t-1}}}{1 - \bar{\alpha}_t}}_{\gamma_0} \boldsymbol{y}_0 + \underbrace{\frac{(1 - \bar{\alpha}_{t-1})\sqrt{\alpha_t}}{1 - \bar{\alpha}_t}}_{\gamma_1} \boldsymbol{y}_t + \underbrace{\left(1 + \frac{(\sqrt{\bar{\alpha}_t} - 1)(\sqrt{\alpha_t} + \sqrt{\bar{\alpha}_{t-1}})}{1 - \bar{\alpha}_t}\right)}_{\gamma_2} f_\phi(\boldsymbol{x}),$$

$$\tilde{\beta}_t := \frac{1 - \bar{\alpha}_{t-1}}{1 - \bar{\alpha}_t}\beta_t.$$

We provide the derivation in Appendix A.1. The labels under the terms are used in Algorithm 2.

## 2.1 CARD for Regression

For regression problems, the goal of the reverse diffusion process is to gradually recover the distribution of the noise term, the aleatoric or local uncertainty inherent in the observations (Kendall and Gal, 2017; Wang and Zhou, 2020), enabling us to generate samples that match the true conditional $p(\boldsymbol{y} \mid \boldsymbol{x})$.

Following the reparameterization introduced by denoising diffusion probabilistic models (DDPM) (Ho et al., 2020), we construct $\epsilon_\theta\big(\boldsymbol{x}, \boldsymbol{y}_t, f_\phi(\boldsymbol{x}), t\big)$, which is a function approximator parameterized by a deep neural network that predicts the forward diffusion noise $\epsilon$ sampled for $\boldsymbol{y}_t$. The training and inference procedure can be carried out in a standard DDPM manner.

---

**Algorithm 1** Training (Regression)
___
1: Pre-train $f_\phi(\boldsymbol{x})$ that predicts $\mathbb{E}(\boldsymbol{y} \mid \boldsymbol{x})$ with MSE
2: **repeat**
3:    Draw $y_0 \sim q(\boldsymbol{y}_0 \mid \boldsymbol{x})$
4:    Draw $t \sim \text{Uniform}(\{1 \ldots T\})$
5:    Draw $\epsilon \sim \mathcal{N}(\boldsymbol{0}, \boldsymbol{I})$
6:    Compute noise estimation loss

$$\mathcal{L}_\epsilon = \big|\big|\epsilon - \epsilon_\theta\big(\boldsymbol{x}, \sqrt{\bar{\alpha}_t}\boldsymbol{y}_0 + \sqrt{1 - \bar{\alpha}_t}\epsilon + (1 - \sqrt{\bar{\alpha}_t})f_\phi(\boldsymbol{x}), f_\phi(\boldsymbol{x}), t\big)\big|\big|^2$$

7:    Take numerical optimization step on:
$$\nabla_\theta \mathcal{L}_\epsilon$$

8: **until** Convergence
___

---
**Algorithm 2** Inference (Regression)
---
1: $\boldsymbol{y}_T \sim \mathcal{N}(f_\phi(\boldsymbol{x}), \boldsymbol{I})$
2: **for** $t = T$ to 1 **do**
3:     Draw $\boldsymbol{z} \sim \mathcal{N}(\boldsymbol{0}, \boldsymbol{I})$ if $t > 1$
4:     Calculate reparameterized $\hat{\boldsymbol{y}}_0 = \frac{1}{\sqrt{\bar{\alpha}_t}} \Big( \boldsymbol{y}_t - (1 - \sqrt{\bar{\alpha}_t}) f_\phi(\boldsymbol{x}) - \sqrt{1 - \bar{\alpha}_t} \boldsymbol{\epsilon}_\theta \big( \boldsymbol{x}, \boldsymbol{y}_t, f_\phi(\boldsymbol{x}), t \big) \Big)$
5:     Let $\boldsymbol{y}_{t-1} = \gamma_0 \hat{\boldsymbol{y}}_0 + \gamma_1 \boldsymbol{y}_t + \gamma_2 f_\phi(\boldsymbol{x}) + \sqrt{\tilde{\beta}_t} \boldsymbol{z}$ if $t > 1$, else set $\boldsymbol{y}_{t-1} = \hat{\boldsymbol{y}}_0$
6: **end for**
7: **return** $\boldsymbol{y}_0$
---

## 2.2 CARD for Classification

We formulate the classification tasks in a similar fashion as in Section 2.1, where we:

1. Replace the continuous response variable with a one-hot encoded label vector for $\boldsymbol{y}_0$;

2. Replace the mean estimator with a pre-trained classifier that outputs softmax probabilities of the class labels for $f_\phi(\boldsymbol{x})$.

This construction no longer assumes $\boldsymbol{y}_0$ to be drawn from a categorical distribution, but instead treats each one-hot label as a class prototype, *i.e.*, we assume a continuous data and state space, which enables us to keep the Gaussian diffusion model framework. The sampling procedure would output reconstructed $\boldsymbol{y}_0$ in the range of real numbers for each dimension, instead of a vector in the probability simplex. Denoting $C$ as the number of classes and $\boldsymbol{1}_C$ as a $C$-dimensional vector of 1s, we convert such output to a probability vector in a softmax form of a temperature-weighted Brier score (Brier, 1950), which computes the squared error between the prediction and $\boldsymbol{1}_C$. Mathematically, the probability of predicting the $k^{th}$ class and the final point prediction $\hat{y}$ can be expressed as

$$\Pr(y = k) = \frac{\exp(-(\boldsymbol{y}_0 - \boldsymbol{1}_C)^2_k/\tau)}{\sum_{i=1}^{C} \exp(-(\boldsymbol{y}_0 - \boldsymbol{1}_C)^2_i/\tau)}; \; \hat{y} = \arg\max_k \big( -(\boldsymbol{y}_0 - \boldsymbol{1}_C)^2_k \big). \tag{10}$$

where $\tau > 0$ is the temperature parameter, and $(\boldsymbol{y}_0 - \boldsymbol{1}_C)^2_k$ indicates the $k^{th}$ dimension of the vector of element-wise square error between $\boldsymbol{y}_0$ and $\boldsymbol{1}_C$, *i.e.*, $(\boldsymbol{y}_0 - \boldsymbol{1}_C)^2_k = \|\boldsymbol{y}_{0_k} - 1\|^2$. Intuitively, this construction would assign the class whose raw output in the sampled $\boldsymbol{y}_0$ is closest to the true class, encoded by the value of 1 in the one-hot label, with the highest probability.

Conditional on the same covariates $\boldsymbol{x}$, the stochasticity of the generative model would give us a different class prototype reconstruction after each reverse process sampling, which enables us to construct predicted probability intervals for all class labels. Such stochastic reconstruction is in a similar fashion as DALL-E 2 (Ramesh et al., 2022) that applies a diffusion prior to reconstruct the image embedding by conditioning on the text embedding during the reverse diffusion process, which is a key step in the diversity of generated images.

# 3 Related Work

Under the supervised learning settings, to model the conditional distribution $p(\boldsymbol{y} \,|\, \boldsymbol{x})$ besides just the conditional mean $\mathbb{E}[\boldsymbol{y} \,|\, \boldsymbol{x}]$ through deep neural networks, existing works have been focusing on quantifying predictive uncertainty, and several lines of work have been proposed. Bayesian neural networks (BNNs) model such uncertainty by assuming distributions over network parameters, capturing the plausibility of the model given the data (Blundell et al., 2015; Hernández-Lobato and Adams, 2015; Gal and Ghahramani, 2016; Kingma et al., 2015; Tomczak et al., 2021). Kendall and Gal (2017) also model the uncertainties in the model outputs besides model parameters, by including the additive noise term as part of the neural network output. Meanwhile, ensemble-based methods (Lakshminarayanan et al., 2017; Liu et al., 2022) have been proposed to model predictive uncertainty by combining multiple neural networks with stochastic outputs. Furthermore, the Neural Processes Family (Garnelo et al., 2018b,a; Kim et al., 2019; Gordon et al., 2020) has introduced a series of models that capture predictive uncertainty in an out-of-distribution fashion, particularly designed for few-shot learning settings.

These above mentioned models have all assumed a parametric form in $p(\boldsymbol{y} \mid \boldsymbol{x})$, namely Gaussian distribution, or a mixture of Gaussians, and optimize the network parameters based on a Gaussian negative log-likelihood objective function. Deep generative models, on the other hand, have been known for modeling implicit distributions without parametric distributional assumptions, but very few works have been proposed to utilize such feature to tackle regression tasks. GAN-based models are introduced by Zhou et al. (2021) and Liu et al. (2021) for conditional density estimation and predictive uncertainty quantification. For classification tasks, on the other hand, generative classifiers (Revow et al., 1996; Fetaya et al., 2020; Ardizzone et al., 2020; Mackowiak et al., 2021) is a class of models that also perform classification with generative models; among them, Zimmermann et al. (2021) propose score-based generative classifiers to tackle classification tasks with score-based generative models (Song et al., 2021b,c). They model $p(\boldsymbol{x} \mid \boldsymbol{y})$ and predict the label with the largest conditional likelihood of $\boldsymbol{x}$, while CARD models $p(\boldsymbol{y} \mid \boldsymbol{x})$ instead.

In recent years, the class of diffusion-based (or score-based) deep generative models has demonstrated its outstanding performance in modeling high-dimensional multi-modal distributions (Ho et al., 2020; Song et al., 2021a; Kawar et al., 2022; Xiao et al., 2022; Dhariwal and Nichol, 2021; Song and Ermon, 2019, 2020), with most work focusing on Gaussian diffusion processes operating in continuous state spaces. Hoogeboom et al. (2021) introduce extensions of diffusion models for categorical data, and Austin et al. (2021) have proposed diffusion models for discrete data as a generalization of the multinomial diffusion models, which could provide an alternative way of performing classification with diffusion-based models.

## 4 Experiments

For the hyperparameters of CARD in both regression and classification tasks, we set the number of timesteps as $T = 1000$, a linear noise schedule with $\beta_1 = 10^{-4}$ and $\beta_T = 0.02$, same as Ho et al. (2020). We provide a more detailed walk-through of the experimental setup, including training and network architecture, in Appendix A.8.

### 4.1 Regression

Putting aside its statistical interpretation, the word *regress* indicates a direction opposite to *progress*, suggesting a less developed state. Such semantics in fact translates well into the statistical domain, in the sense that traditional regression analysis methods often only focus on estimating $\mathbb{E}(\boldsymbol{y} \mid \boldsymbol{x})$, while leaving out all remaining details about $p(\boldsymbol{y} \mid \boldsymbol{x})$. In recent years, Bayesian neural networks (BNNs) have emerged as a class of models that aims at estimating the uncertainty (Hernández-Lobato and Adams, 2015; Gal and Ghahramani, 2016; Lakshminarayanan et al., 2017; Tomczak et al., 2021), providing a more complete picture of $p(\boldsymbol{y} \mid \boldsymbol{x})$. The metric that they use to quantify uncertainty estimation, negative log-likelihood (NLL), is computed with a Gaussian density, implying their assumption such that the conditional distributions $p(\boldsymbol{y} \mid \boldsymbol{x} = x)$ for all $x$ are Gaussian. However, this assumption is very difficult to verify for real-world datasets: the covariates can be arbitrarily high-dimensional, making the feature space increasingly sparse with respect to the number of collected observations.

To accommodate the need for uncertainty estimation without imposing such restriction for the parametric form of $p(\boldsymbol{y} \mid \boldsymbol{x})$, we apply the following two metrics, both of which are designed to empirically evaluate the level of similarity between the learned and the true conditional distributions:

1. Prediction Interval Coverage Probability (PICP);
2. Quantile Interval Coverage Error (QICE).

PICP has been described in Yao et al. (2019), whereas QICE is a new metric proposed by us. We describe both of them in what follows.

### 4.1.1 PICP and QICE

The PICP is computed as

$$\text{PICP} := \frac{1}{N} \sum_{n=1}^{N} \mathbb{1}_{y_n \geq \hat{y}_n^{\text{low}}} \cdot \mathbb{1}_{y_n \leq \hat{y}_n^{\text{high}}}, \tag{11}$$

where $\hat{y}_n^{\text{low}}$ and $\hat{y}_n^{\text{high}}$ represent the low and high percentiles, respectively, of our choice for the predicted $y$ outputs given the same $x$ input. This metric measures the proportion of true observations that fall in the percentile range of the generated $y$ samples given each $x$ input. Intuitively, when the learned distribution represents the true distribution well, this measurement should be close to the difference between the selected low and high percentiles. In this paper, we choose the $2.5^{th}$ and $97.5^{th}$ percentile, thus an ideal PICP value for the learned model should be 95%.

Meanwhile, there is a caveat for this metric: for example, imagine a situation where the $2.5^{th}$ to $97.5^{th}$ percentile of the learned distribution happens to cover the data between the $1^{st}$ and $96^{th}$ percentiles from the true distribution. Given enough samples, we shall still obtain a PICP value close to 95%, but clearly there is a mismatch between the learned distribution and the true one.

Based on such reasoning, we propose a new empirical metric QICE, which by design can be viewed as PICP with finer granularity, and without uncovered quantile ranges. To compute QICE, we first generate enough $y$ samples given each $x$, and divide them into $M$ bins with roughly equal sizes. We would obtain the corresponding quantile values at each boundary. In this paper, we set $M = 10$, and obtain the following 10 quantile intervals (QIs) of the generated $y$ samples: below the $10^{th}$ percentile, between the $10^{th}$ and $20^{th}$ percentiles, ..., between the $80^{th}$ and $90^{th}$ percentiles, and above the $90^{th}$ percentile. Optimally, when the learned conditional distribution is identical to the true one, given enough samples from both learned and true distribution we shall observe about 10% of true data falling into each of these 10 QIs.

We define *QICE* to be the mean absolute error between the proportion of true data contained by each QI and the optimal proportion, which is $1/M$ for all intervals:

$$\text{QICE} := \frac{1}{M} \sum_{m=1}^{M} \left| r_m - \frac{1}{M} \right|, \text{ where } r_m = \frac{1}{N} \sum_{n=1}^{N} \mathbb{1}_{y_n \geq \hat{y}_n^{\text{low}_m}} \cdot \mathbb{1}_{y_n \leq \hat{y}_n^{\text{high}_m}}. \tag{12}$$

Intuitively, under optimal scenario with enough samples, we shall obtain a QICE value of 0. Note that each $r_m$ is indeed the PICP for the corresponding QI with boundaries at $\hat{y}_n^{\text{low}_m}$ and $\hat{y}_n^{\text{high}_m}$. Since the true $y$ for each $x$ is guaranteed to fall into one of these QIs, we are thus able to overcome the mismatch issue described in the above example for PICP: fewer true instances falling into one QI would result in more instances captured by another QI, thus increasing the absolute error for both QIs.

QICE is similar to NLL in the sense that it also utilizes the summary statistics of the samples from the learned distribution conditional on each new $x$ to empirically evaluate how well the model fits the true data. Meanwhile, it does not assume any parametric form on the conditional distribution, making it a much more generalizable metric to measure the level of distributional match between the learned and the underlying true conditional distributions, especially when the true conditional distribution is known to be multi-modal. We will demonstrate this point through the regression toy examples.

### 4.1.2 Toy Examples

To demonstrate the effectiveness of CARD in regression tasks for not only learning the conditional mean $\mathbb{E}(y \,|\, x)$, but also recreating the ground truth data generating mechanism, we first apply CARD on 8 toy examples, whose data generating functions are designed to possess different statistical characteristics: some have a uni-modal symmetric distribution for their error term (linear regression, quadratic regression, sinusoidal regression), others have heteroscedasticity (log-log linear regression, log-log cubic regression) or multi-modality (inverse sinusoidal regression, 8 Gaussians, full circle). We show that the trained CARD models can generate samples that are visually indistinguishable from the true response variables of the new covariates, as well as quantitatively match the true distribution in terms of some summary statistics. We present the scatter plots of both true and generated data for all 8 tasks in Figure 1. For tasks with uni-modal conditional distribution, we fill the region between the $2.5^{th}$ and $97.5^{th}$ percentile of the generated $y$'s. We observe that within each task, the generated samples blend remarkably well with the true test instances, suggesting the capability of reconstructing the underlying data generation mechanism by CARD. A more detailed description of the toy examples, including more quantitative analyses, is presented in Appendix A.13.

### 4.1.3 UCI Regression Tasks

We continue to investigate our model through experiments on real-world datasets. We adopt the same set of 10 UCI regression benchmark datasets (Dua and Graff, 2017) as well as the experimental

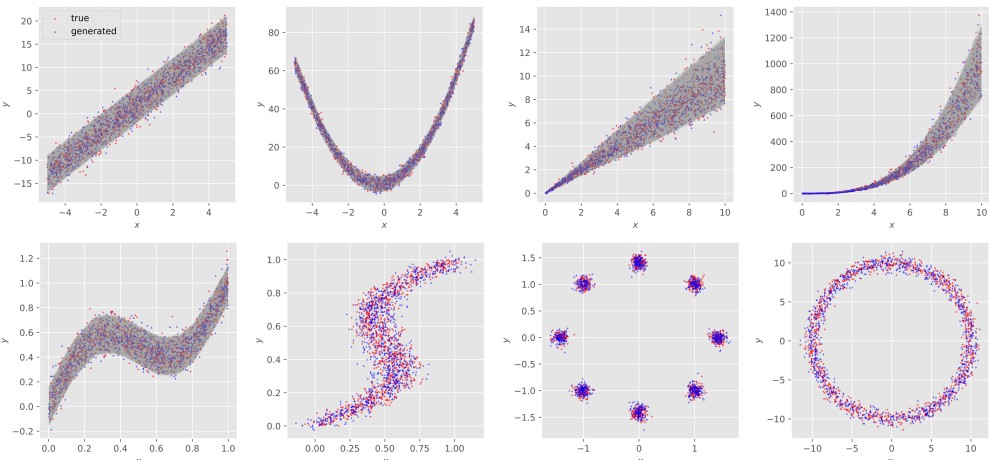

Figure 1: Regression toy example scatter plots. (**Top**) left to right: linear regression, quadratic regression, log-log linear regression, log-log cubic regression; (**Bottom**) left to right: sinusoidal regression, inverse sinusoidal regression, 8 Gaussians, full circle.

protocol proposed by Hernández-Lobato and Adams (2015) and followed by Gal and Ghahramani (2016) and Lakshminarayanan et al. (2017). The dataset information is provided in Table 14.

We apply multiple train-test splits with $90\%/10\%$ ratio in the same way as Hernández-Lobato and Adams (2015) (20 folds for all datasets except 5 for Protein and 1 for Year), and report the metrics by their mean and standard deviation across all splits. We compare our method to all aforementioned BNN frameworks: PBP, MC Dropout, and Deep Ensembles, as well as another deep generative model that estimates a conditional distribution sampler, GCDS (Zhou et al., 2021). We note that GCDS is related to a concurrent work of Yang et al. (2022), who share a comparable idea but use it in a different application. Following the same paradigm of BNN model assessment, we evaluate the accuracy and predictive uncertainty estimation of CARD by reporting RMSE and NLL. Furthermore, we also report QICE for all methods to evaluate distributional matching. Since this new metric was not applied in previous methods, we re-ran the experiments for all BNNs and obtained comparable or slightly better results in terms of other commonly used metrics reported in their literature. Further details about the experimental setup for these models can be found in Appendix A.9. The experiment results with corresponding metrics are shown in Tables 1, 2, and 3, with the number of times that each model achieves the best corresponding metric reported in the last row.

We observe that CARD outperforms existing methods, often by a considerable margin (especially on larger datasets), in all metrics for most of the datasets, and is competitive with the best method for the remaining ones: we obtain state-of-the-art results in 9 out of 10 datasets in terms of RMSE, 8 out of 10 for NLL, and 5 out of 10 for QICE. It is worth noting that although we do not explicitly optimize our model by MSE or by NLL, we still obtain better results than models trained with these objectives.

## 4.2 Classification

Similar to Lakshminarayanan et al. (2017), our motivation for classification is not to achieve state-of-the-art performance in terms of mean accuracy on the benchmark datasets, which is strongly related to network architecture design. Our goal is two-fold:

1. We aim to solve classification problems via a generative model, emphasizing its capability to improve the performance of a base classifier with deterministic outputs in terms of accuracy;

2. We intend to provide an alternative sense of uncertainty, by introducing the idea of model confidence at the instance level, *i.e.*, how sure the model is about *each* of its predictions, through the stochasticity of outputs from a generative model.

As another type of supervised learning problems, classification is different from regression mainly for the response variable being discrete class labels instead of continuous values. The conventional operation is to cast the classifier output as a point estimate, with a value between $0$ and $1$. Such

Table 1: RMSE of UCI regression tasks. For both [1]Kin8nm and [2]Naval dataset, we multiply the response variable by 100 to match the scale of others.

| Dataset | RMSE ↓ | | | | |
| | PBP | MC Dropout | Deep Ensembles | GCDS | CARD (ours) |
|---|---|---|---|---|---|
| Boston | $2.89 \pm 0.74$ | $3.06 \pm 0.96$ | $3.17 \pm 1.05$ | $2.75 \pm 0.58$ | $\mathbf{2.61 \pm 0.63}$ |
| Concrete | $5.55 \pm 0.46$ | $5.09 \pm 0.60$ | $4.91 \pm 0.47$ | $5.39 \pm 0.55$ | $\mathbf{4.77 \pm 0.46}$ |
| Energy | $1.58 \pm 0.21$ | $1.70 \pm 0.22$ | $2.02 \pm 0.32$ | $0.64 \pm 0.09$ | $\mathbf{0.52 \pm 0.07}$ |
| Kin8nm[1] | $9.42 \pm 0.29$ | $7.10 \pm 0.26$ | $8.65 \pm 0.47$ | $8.88 \pm 0.42$ | $\mathbf{6.32 \pm 0.18}$ |
| Naval[2] | $0.41 \pm 0.08$ | $0.08 \pm 0.03$ | $0.09 \pm 0.01$ | $0.14 \pm 0.05$ | $\mathbf{0.02 \pm 0.00}$ |
| Power | $4.10 \pm 0.15$ | $4.04 \pm 0.14$ | $4.02 \pm 0.15$ | $4.11 \pm 0.16$ | $\mathbf{3.93 \pm 0.17}$ |
| Protein | $4.65 \pm 0.02$ | $4.16 \pm 0.12$ | $4.45 \pm 0.02$ | $4.50 \pm 0.02$ | $\mathbf{3.73 \pm 0.01}$ |
| Wine | $0.64 \pm 0.04$ | $\mathbf{0.62 \pm 0.04}$ | $0.63 \pm 0.04$ | $0.66 \pm 0.04$ | $0.63 \pm 0.04$ |
| Yacht | $0.88 \pm 0.22$ | $0.84 \pm 0.27$ | $1.19 \pm 0.49$ | $0.79 \pm 0.26$ | $\mathbf{0.65 \pm 0.25}$ |
| Year | $8.86 \pm$ NA | $8.77 \pm$ NA | $8.79 \pm$ NA | $9.20 \pm$ NA | $\mathbf{8.70 \pm}$ NA |
| # best | 0 | 1 | 0 | 0 | **9** |

Table 2: NLL of UCI regression tasks.

| Dataset | NLL ↓ | | | | |
| | PBP | MC Dropout | Deep Ensembles | GCDS | CARD (ours) |
|---|---|---|---|---|---|
| Boston | $2.53 \pm 0.27$ | $2.46 \pm 0.12$ | $2.35 \pm 0.16$ | $18.66 \pm 8.92$ | $\mathbf{2.35 \pm 0.12}$ |
| Concrete | $3.19 \pm 0.05$ | $3.21 \pm 0.18$ | $\mathbf{2.93 \pm 0.12}$ | $13.64 \pm 6.88$ | $2.96 \pm 0.09$ |
| Energy | $2.05 \pm 0.05$ | $1.50 \pm 0.11$ | $1.40 \pm 0.27$ | $1.46 \pm 0.72$ | $\mathbf{1.04 \pm 0.06}$ |
| Kin8nm | $-0.83 \pm 0.02$ | $-1.14 \pm 0.05$ | $-1.06 \pm 0.02$ | $-0.38 \pm 0.36$ | $\mathbf{-1.32 \pm 0.02}$ |
| Naval | $-3.97 \pm 0.10$ | $-4.45 \pm 0.38$ | $-5.94 \pm 0.10$ | $-5.06 \pm 0.48$ | $\mathbf{-7.54 \pm 0.05}$ |
| Power | $2.92 \pm 0.02$ | $2.90 \pm 0.03$ | $2.89 \pm 0.02$ | $2.83 \pm 0.06$ | $\mathbf{2.82 \pm 0.02}$ |
| Protein | $3.05 \pm 0.00$ | $2.80 \pm 0.08$ | $2.89 \pm 0.02$ | $2.81 \pm 0.09$ | $\mathbf{2.49 \pm 0.03}$ |
| Wine | $1.03 \pm 0.03$ | $0.93 \pm 0.06$ | $0.96 \pm 0.06$ | $6.52 \pm 21.86$ | $\mathbf{0.92 \pm 0.05}$ |
| Yacht | $1.58 \pm 0.08$ | $1.73 \pm 0.22$ | $1.11 \pm 0.18$ | $\mathbf{0.61 \pm 0.34}$ | $0.90 \pm 0.08$ |
| Year | $3.69 \pm$ NA | $3.42 \pm$ NA | $3.44 \pm$ NA | $3.43 \pm$ NA | $\mathbf{3.34 \pm}$ NA |
| # best | 0 | 0 | 1 | 1 | **8** |

Table 3: QICE (in %) of UCI regression tasks.

| Dataset | QICE ↓ | | | | |
| | PBP | MC Dropout | Deep Ensembles | GCDS | CARD (ours) |
|---|---|---|---|---|---|
| Boston | $3.50 \pm 0.88$ | $3.82 \pm 0.82$ | $\mathbf{3.37 \pm 0.00}$ | $11.73 \pm 1.05$ | $3.45 \pm 0.83$ |
| Concrete | $2.52 \pm 0.60$ | $4.17 \pm 1.06$ | $2.68 \pm 0.64$ | $10.49 \pm 1.01$ | $\mathbf{2.30 \pm 0.66}$ |
| Energy | $6.54 \pm 0.90$ | $5.22 \pm 1.02$ | $\mathbf{3.62 \pm 0.58}$ | $7.41 \pm 2.19$ | $4.91 \pm 0.94$ |
| Kin8nm | $1.31 \pm 0.25$ | $1.50 \pm 0.32$ | $1.17 \pm 0.22$ | $7.73 \pm 0.80$ | $\mathbf{0.92 \pm 0.25}$ |
| Naval | $4.06 \pm 1.25$ | $12.50 \pm 1.95$ | $6.64 \pm 0.60$ | $5.76 \pm 2.25$ | $\mathbf{0.80 \pm 0.21}$ |
| Power | $\mathbf{0.82 \pm 0.19}$ | $1.32 \pm 0.37$ | $1.09 \pm 0.26$ | $1.77 \pm 0.33$ | $0.92 \pm 0.21$ |
| Protein | $1.69 \pm 0.09$ | $2.82 \pm 0.41$ | $2.17 \pm 0.16$ | $2.33 \pm 0.18$ | $\mathbf{0.71 \pm 0.11}$ |
| Wine | $\mathbf{2.22 \pm 0.64}$ | $2.79 \pm 0.56$ | $2.37 \pm 0.63$ | $3.13 \pm 0.79$ | $3.39 \pm 0.45$ |
| Yacht | $6.93 \pm 1.74$ | $10.33 \pm 1.34$ | $7.22 \pm 1.41$ | $\mathbf{5.01 \pm 1.02}$ | $8.03 \pm 1.17$ |
| Year | $2.96 \pm$ NA | $2.43 \pm$ NA | $2.56 \pm$ NA | $1.61 \pm$ NA | $\mathbf{0.53 \pm}$ NA |
| # best | 2 | 0 | 2 | 1 | **5** |

design is intended for prediction interpretability: since humans already have a cognitive intuition for probabilities (Cosmides and Tooby, 1996), the output from a classification model is intended to convey a sense of likelihood for a particular class label. In other words, the predicted probability should reflect its confidence, *i.e.*, a level of certainty, in predicting such a label. Guo et al. (2017) provide the following example of a good classifier, whose output aligns with human intuition for probabilities: if the model outputs a probability prediction of $0.8$, we hope it indicates that the model is 80% sure that its prediction is correct; given 100 predictions of $0.8$, one shall expect roughly 80 of them are correct.

In that sense, a good classification algorithm not only can predict the correct label, but also can reflect the true correctness likelihood through its probability predictions, *i.e.*, providing calibrated confidence (Guo et al., 2017). To evaluate the level of miscalibration by a model, metrics like Expected Calibration Error (ECE) and Maximum Calibration Error (MCE) (Naeini et al., 2015) have been adopted in recent literature (Kristiadi et al., 2022; Rudner et al., 2021) for image classification tasks, and calibration methods like Platt scaling and isotonic regression have been developed to improve such alignment (Guo et al., 2017).

Note that these methods are all based on point estimate predictions by the classifier. Furthermore, these alignment metrics can only be computed at a subgroup level in practice, instead of at the instance level. In other words, one may not be able to make the claim with the existing classification

framework such that *given a particular test instance*, how confident the classifier is in its prediction to be correct. We discuss our analysis in ECE with more details in Appendix A.16, which may help justify our motivation in introducing an alternative way to measure model confidence at the level of *individual test instances* in Section 4.2.1.

### 4.2.1 Predict with Instance Level Model Confidence via Generative Models

We propose the following framework to assess model confidence for its predictions *at the instance level*: for each test instance, we first sample $N$ class prototype reconstructions by CARD through the classification version of Algorithm 2, and then perform the following computations:

1. We directly calculate the prediction interval width (PIW) between the $2.5^{th}$ and $97.5^{th}$ percentiles of the $N$ reconstructed values for all classes, *i.e.*, with $C$ different classes in total, we would obtain $C$ PIWs for each instance;

2. We then convert the samples into probability space with Eq. (10), and apply paired two-sample $t$-test as an uncertainty estimation method proposed in Fan et al. (2021): we obtain the most and second most predicted classes for each instance, and test whether the difference in their mean predicted probability is statistically significant.

This framework would require the classifier to not produce the exact same output each time, since the goal is to construct prediction intervals for each of the class labels. Therefore, the class of generative models is a preferable modeling choice due to its ability to produce stochastic outputs, instead of just a point estimate by traditional classifiers.

In practice, we view each one-hot label as a class prototype in real continuous space (introduced in Section 2.2), and we use a generative model to reconstruct this prototype in a stochastic fashion. The intuition is that if the classifier is sure about the class that a particular instance belongs to, it would precisely reconstruct the original prototype vector without much uncertainty; otherwise, different class prototype reconstructions of the same test instance tend to have more variations: under the context of denoising diffusion models, given different samples from the prior distribution at timestep $T$, the label reconstructions would appear rather different from each other.

### 4.2.2 Classification with Model Confidence on CIFAR-10 Dataset

We demonstrate our experimental results on the CIFAR-10 dataset. We first contextualize the performance of CARD in conventional metrics including accuracy and NLL with other BNNs in ResNet-18 architecture in Table 4. The metrics of other methods were reported in Tomczak et al. (2021), a recent work in BNNs that proposes tighter ELBOs to improve variational inference performance and prior hyperparameter optimization. Following the recipe in Section 2.2, we first pre-train a deterministic classifier with the same ResNet-18 architecture, and achieve a test accuracy of $90.39\%$, with which we proceed to train CARD. We then obtain our instance prediction through majority vote, *i.e.*, the most predicted class label among its $N$ samples for each image input, and achieve an improved test accuracy with a mean of $90.93\%$ across 10 runs, showing its ability to improve test accuracy from the base classifier. Our NLL result is competitive among the best ones, even though the model is not optimized with a cross-entropy objective function, as we assume the class labels to be in the real continuous space.

Table 4: Comparison of accuracy (in %) and NLL for CIFAR-10 classification with other BNNs.

| Model | CMV-MF-VI | CM-MF-VI | CV-MF-VI | MF-VI | MC Dropout | MAP | CARD |
|---|---|---|---|---|---|---|---|
| Accuracy | $86.25 \pm 0.06$ | $86.66 \pm 0.24$ | $79.78 \pm 0.30$ | $77.08 \pm 1.14$ | $83.64 \pm 0.28$ | $84.69 \pm 0.35$ | $\mathbf{90.93 \pm 0.02}$ |
| NLL | $0.41 \pm 0.00$ | $\mathbf{0.39 \pm 0.00}$ | $0.59 \pm 0.00$ | $0.68 \pm 0.02$ | $0.49 \pm 0.00$ | $0.93 \pm 0.02$ | $0.46 \pm 0.00$ |

We now present the results from one model run with the proposed framework in Section 4.2.1 for evaluating the instance-level prediction confidence. After obtaining the PIW and paired two-sample $t$-test ($\alpha = 0.05$) result from each test instance, we first split the test instances into two groups by the correctness of majority-vote predictions, then we obtain only the PIW corresponding to the true class for each instance, and compute the mean PIW of the true class within each group. In addition, we split the test instances by $t$-test rejection status, and compute the mean accuracy in each group. We

report the results from these two grouping procedures in Table 5, where the metrics are computed across all test instances and at the level of each true class label.

Table 5: PIW (multiplied by $100$) and $t$-test results for CIFAR-10 classification task.

| Class | Accuracy | PIW | | Accuracy by $t$-test Status | |
|---|---|---|---|---|---|
| | | Correct | Incorrect | Rejected | Not-Rejected (Count) |
| All | 90.95% | 2.37 | 21.52 | 91.25% | 42.86% (63) |
| 1 | 91.00% | 3.28 | 18.83 | 91.51% | 45.45% (11) |
| 2 | 96.00% | 0.55 | 29.27 | 96.19% | 33.33% (3) |
| 3 | 87.30% | 2.65 | 24.40 | 87.55% | 25.00% (4) |
| 4 | 81.90% | 5.48 | 21.45 | 82.10% | 63.64% (11) |
| 5 | 93.30% | 2.41 | 30.02 | 93.67% | 20.00% (5) |
| 6 | 84.70% | 4.16 | 19.57 | 85.21% | 46.15% (13) |
| 7 | 94.20% | 1.84 | 26.01 | 94.38% | 33.33% (3) |
| 8 | 92.80% | 1.96 | 19.35 | 93.07% | 25.00% (4) |
| 9 | 95.30% | 0.56 | 15.75 | 95.49% | 33.33% (3) |
| 10 | 93.00% | 1.50 | 14.04 | 93.26% | 50.00% (6) |

We observe from Table 5 that under the scope of the entire test set, the mean PIW of the true class label among the correct predictions is narrower than that of the incorrect predictions by an order of magnitude, indicating that when CARD is making correct predictions, its class label reconstructions have much smaller variations. We may interpret such results as that CARD can reveal what it does not know through the relativity in reconstruction variations. Furthermore, when comparing the mean PIWs across different classes, we observe that the class with a higher prediction accuracy tends to have a sharper contrast in true label PIW between correct and incorrect predictions; additionally, the PIW values of both correct and incorrect predictions tend to be larger in a less accurate class. Meanwhile, it is worth noting that if we predict the class label by the one with the narrowest PIW for each instance, we can already obtain a test accuracy of $87.84\%$, suggesting a strong correlation between the prediction correctness and instance-level model confidence (in terms of label reconstruction variability). Moreover, we observe that the accuracy of test instances rejected by the $t$-test is much higher than that of the not-rejected ones, both across the entire test set and within each class.

We point out that these metrics can reflect how sure CARD is about the correctness of its predictions, and can thus be used as an important indicator of whether the model prediction *of each instance* can be trusted or not. Therefore, it has the potential to be further applied in the human-machine collaboration domain (Madras et al., 2018; Raghu et al., 2019; Wilder et al., 2020; Gao et al., 2021), such that one can apply such uncertainty measurement to decide if we can directly accept the model prediction, or we need to allocate the instance to humans for further evaluation.

## 5 Conclusion

In this paper, we propose Classification And Regression Diffusion (CARD) models, a class of conditional generative models that approaches supervised learning problems from a conditional generation perspective. Without training with objectives directly related to the evaluation metrics, we achieve state-of-the-art results on benchmark regression tasks. Furthermore, CARD exhibits a strong ability to represent the conditional distribution with multiple density modes. We also propose a new metric Quantile Interval Coverage Error (QICE), which can be viewed as a generalized version of negative log-likelihood in evaluating how well the model fits the data. Lastly, we introduce a framework to evaluate prediction uncertainty at the instance level for classification tasks.

## Acknowledgments

The authors acknowledge the support of NSF IIS 1812699 and 2212418, and the Texas Advanced Computing Center (TACC) for providing HPC resources that have contributed to the research results reported within this paper.

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
