# A Derivations and Additional Experiment Results

## A.1 Derivations

**Derivation for Forward Process Posteriors:** In this section, we derive the mean and variance of the forward process posteriors $q(\boldsymbol{y}_{t-1} \,|\, \boldsymbol{y}_t, \boldsymbol{y}_0, \boldsymbol{x})$ in Eq. (9), which are tractable when conditioned on $\boldsymbol{y}_0$:

$$q(\boldsymbol{y}_{t-1} \,|\, \boldsymbol{y}_t, \boldsymbol{y}_0, \boldsymbol{x}) \propto \underbrace{q\big(\boldsymbol{y}_t \,|\, \boldsymbol{y}_{t-1}, f_\phi(\boldsymbol{x})\big)}_{\text{Eq. (7)}} \underbrace{q\big(\boldsymbol{y}_{t-1} \,|\, \boldsymbol{y}_0, f_\phi(\boldsymbol{x})\big)}_{\text{Eq. (8)}} \tag{13}$$

$$\propto \exp\Bigg(-\frac{1}{2}\Bigg(\frac{\big(\boldsymbol{y}_t - (1-\sqrt{\alpha_t})f_\phi(\boldsymbol{x}) - \sqrt{\alpha_t}\boldsymbol{y}_{t-1}\big)^2}{\beta_t}$$
$$+\frac{\big(\boldsymbol{y}_{t-1} - \sqrt{\bar{\alpha}_{t-1}}\boldsymbol{y}_0 - (1-\sqrt{\bar{\alpha}_{t-1}})f_\phi(\boldsymbol{x})\big)^2}{1-\bar{\alpha}_{t-1}}\Bigg)\Bigg) \tag{14}$$

$$\propto \exp\Bigg(-\frac{1}{2}\Bigg(\frac{\alpha_t \boldsymbol{y}_{t-1}^2 - 2\sqrt{\alpha_t}\big(\boldsymbol{y}_t - (1-\sqrt{\alpha_t})f_\phi(\boldsymbol{x})\big)\boldsymbol{y}_{t-1}}{\beta_t}$$
$$+\frac{\boldsymbol{y}_{t-1}^2 - 2\big(\sqrt{\bar{\alpha}_{t-1}}\boldsymbol{y}_0 + (1-\sqrt{\bar{\alpha}_{t-1}})f_\phi(\boldsymbol{x})\big)\boldsymbol{y}_{t-1}}{1-\bar{\alpha}_{t-1}}\Bigg)\Bigg) \tag{15}$$

$$= \exp\Bigg(-\frac{1}{2}\Bigg(\underbrace{\Big(\frac{\alpha_t}{\beta_t} + \frac{1}{1-\bar{\alpha}_{t-1}}\Big)}_{\textcircled{1}}\boldsymbol{y}_{t-1}^2$$
$$-2\underbrace{\Big(\frac{\sqrt{\bar{\alpha}_{t-1}}}{1-\bar{\alpha}_{t-1}}\boldsymbol{y}_0 + \frac{\sqrt{\alpha_t}}{\beta_t}\boldsymbol{y}_t + \Big(\frac{\sqrt{\alpha_t}(\sqrt{\alpha_t}-1)}{\beta_t} + \frac{1-\sqrt{\bar{\alpha}_{t-1}}}{1-\bar{\alpha}_{t-1}}\Big)f_\phi(\boldsymbol{x})\Big)}_{\textcircled{2}}\boldsymbol{y}_{t-1}\Bigg)\Bigg), \tag{16}$$

where

$$\textcircled{1} = \frac{\alpha_t(1-\bar{\alpha}_{t-1}) + \beta_t}{\beta_t(1-\bar{\alpha}_{t-1})} = \frac{1-\bar{\alpha}_t}{\beta_t(1-\bar{\alpha}_{t-1})}, \tag{17}$$

and we have the posterior variance

$$\tilde{\beta}_t = \frac{1}{\textcircled{1}} = \frac{1-\bar{\alpha}_{t-1}}{1-\bar{\alpha}_t}\beta_t. \tag{18}$$

Meanwhile, the following coefficients of the terms in the posterior mean through dividing each coefficient in $\textcircled{2}$ by $\textcircled{1}$:

$$\gamma_0 = \frac{\sqrt{\bar{\alpha}_{t-1}}}{1-\bar{\alpha}_{t-1}}\Big/\textcircled{1} = \frac{\sqrt{\bar{\alpha}_{t-1}}}{1-\bar{\alpha}_t}\beta_t, \tag{19}$$

$$\gamma_1 = \frac{\sqrt{\alpha_t}}{\beta_t}\Big/\textcircled{1} = \frac{1-\bar{\alpha}_{t-1}}{1-\bar{\alpha}_t}\sqrt{\alpha_t}, \tag{20}$$

and

$$\gamma_2 = \Big(\frac{\sqrt{\alpha_t}(\sqrt{\alpha_t}-1)}{\beta_t} + \frac{1-\sqrt{\bar{\alpha}_{t-1}}}{1-\bar{\alpha}_{t-1}}\Big)\Big/\textcircled{1}$$
$$= \frac{\alpha_t - \bar{\alpha}_t - \sqrt{\alpha_t}(1-\bar{\alpha}_{t-1}) + \beta_t - \beta_t\sqrt{\bar{\alpha}_{t-1}}}{1-\bar{\alpha}_t}$$
$$= 1 + \frac{(\sqrt{\alpha_t}-1)(\sqrt{\alpha_t} + \sqrt{\bar{\alpha}_{t-1}})}{1-\bar{\alpha}_t}, \tag{21}$$

which together give us the posterior mean

$$\tilde{\mu}\big(\boldsymbol{y}_t, \boldsymbol{y}_0, f_\phi(\boldsymbol{x})\big) = \gamma_0\boldsymbol{y}_0 + \gamma_1\boldsymbol{y}_t + \gamma_2 f_\phi(\boldsymbol{x}).$$

**Derivation for Forward Process Sampling Distribution with Arbitrary Timesteps:** For completeness, we include the derivation for the parameters of the forward diffusion process sampling distribution with arbitrary $t$ steps.

The expectation term is based on Eqs. (36)–(38) in Pandey et al. (2022). From Eq. (7), we have that for all $t = 1, \ldots, T$,

$$\boldsymbol{y}_t = \sqrt{1 - \beta_t}\boldsymbol{y}_{t-1} + (1 - \sqrt{1 - \beta_t})f_\phi(\boldsymbol{x}) + \sqrt{\beta_t}\boldsymbol{\epsilon}, \text{ where } \boldsymbol{\epsilon} \sim \mathcal{N}(\boldsymbol{0}, \boldsymbol{I}). \tag{22}$$

Taking expectation of both sides, we have

$$\mathbb{E}(\boldsymbol{y}_t) = \sqrt{1 - \beta_t}\mathbb{E}(\boldsymbol{y}_{t-1}) + (1 - \sqrt{1 - \beta_t})f_\phi(\boldsymbol{x}) \tag{23}$$

$$= \sqrt{1 - \beta_t}\big(\sqrt{1 - \beta_{t-1}}\mathbb{E}(\boldsymbol{y}_{t-2}) + (1 - \sqrt{1 - \beta_{t-1}})f_\phi(\boldsymbol{x})\big) + (1 - \sqrt{1 - \beta_t})f_\phi(\boldsymbol{x})$$

$$= \sqrt{(1 - \beta_t)(1 - \beta_{t-1})}\mathbb{E}(\boldsymbol{y}_{t-2}) + \big(1 - \sqrt{(1 - \beta_t)(1 - \beta_{t-1})}\big)f_\phi(\boldsymbol{x}) \tag{24}$$

$$\vdots$$

$$= \sqrt{\prod_{i=2}^{t}(1 - \beta_i)}\mathbb{E}(\boldsymbol{y}_1) + \left(1 - \sqrt{\prod_{i=2}^{t}(1 - \beta_i)}\right)f_\phi(\boldsymbol{x}) \tag{25}$$

$$= \sqrt{\prod_{i=2}^{t}(1 - \beta_i)}\big(\sqrt{1 - \beta_1}\boldsymbol{y}_0 + (1 - \sqrt{1 - \beta_1})f_\phi(\boldsymbol{x})\big) + \left(1 - \sqrt{\prod_{i=2}^{t}(1 - \beta_i)}\right)f_\phi(\boldsymbol{x}) \tag{26}$$

$$= \sqrt{\prod_{i=1}^{t}(1 - \beta_i)}\boldsymbol{y}_0 + \left(1 - \sqrt{\prod_{i=1}^{t}(1 - \beta_i)}\right)f_\phi(\boldsymbol{x}) \tag{27}$$

$$= \sqrt{\bar{\alpha}_t}\boldsymbol{y}_0 + (1 - \sqrt{\bar{\alpha}_t})f_\phi(\boldsymbol{x}). \tag{28}$$

Meanwhile, since the addition of two independent Gaussians with different variances, $\mathcal{N}(\boldsymbol{0}, \sigma_a^2\boldsymbol{I})$ and $\mathcal{N}(\boldsymbol{0}, \sigma_b^2\boldsymbol{I})$, is distributed as $\mathcal{N}(\boldsymbol{0}, (\sigma_a^2 + \sigma_b^2)\boldsymbol{I})$, we can derive the variance term accordingly,

$$\sigma^2(\boldsymbol{y}_t) = (1 - \bar{\alpha}_t)\boldsymbol{I}. \tag{29}$$

## A.2 More In-Depth Discussion on Several Related Works

In this section, we discuss the relations between CARD and several existing works, which are briefly addressed in Section 3, in more depth.

### A.2.1 Comparing CARD with the Neural Processes Family

In short, CARD models $p(\boldsymbol{y} \mid \boldsymbol{x}, \mathcal{D}_{\text{in}})$, while the Neural Processes Family (NPF) (Garnelo et al., 2018b,a; Kim et al., 2019; Gordon et al., 2020) models $p(\boldsymbol{y} \mid \boldsymbol{x}, \mathcal{D}_{\text{out}})$, where $\mathcal{D}_{\text{in}}$ and $\mathcal{D}_{\text{out}}$ represents in-distribution dataset and out-of-distribution dataset, respectively.

Although both classes of methods can be expressed as modeling $p(\boldsymbol{y} \mid \boldsymbol{x}, \mathcal{D})$, CARD assumes such $(\boldsymbol{x}, \boldsymbol{y})$ comes from the same data-generating mechanism as the set $\mathcal{D}$, while NPF assumes $(\boldsymbol{x}, \boldsymbol{y})$ to be not from the same distribution as $\mathcal{D}$. While CARD fits in the traditional supervised learning setting for in-distribution generalization, NPF is specifically suited for few-shot learning scenarios, where a good model would capture enough pattern from previously seen datasets so that it can generalize well with very limited samples from the new dataset.

Furthermore, both classes of models are capable of generating stochastic outputs, where CARD aims to capture aleatoric uncertainty, which is intrinsic to the data (thus cannot be reduced), while NPF can express epistemic uncertainty as it proposes more diverse functional forms at regions where data is sparse (and such uncertainty would reduce when more data is given). In terms of the conditioning of $\mathcal{D}$, the information of $\mathcal{D}$ is amortized into the network $\epsilon_\theta$ for CARD, while it is included as an explicit representation in the network of NPF that outputs the distribution parameters for $p(\boldsymbol{y} \mid \boldsymbol{x})$. It is also

worth pointing out that CARD does not assume a parametric distributional form for $p(\boldsymbol{y} \mid \boldsymbol{x}, \mathcal{D})$, while NPF assumes a Gaussian distribution, and designs the objective function with such assumption.

The concept and comparison between epistemic and aleatoric uncertainty is more thoroughly discussed by Kendall and Gal (2017), in which we quote, "Out-of-data examples, which can be identified with epistemic uncertainty, cannot be identified with aleatoric uncertainty alone." We acknowledge that modeling OOD uncertainty is an important topic for regression tasks; however, we design our model to focus on modeling aleatoric uncertainty in this paper. We plan to explore CARD's ability in extrapolation as part of our future work.

### A.2.2 Comparing CARD with Discrete Diffusion Models

To construct our framework for classification, we assume the class labels (in terms of one-hot vectors) are from real continuous spaces instead of discrete ones. This assumption enables us to model the forward diffusion process and prior distribution at timestep $T$ with Gaussian distributions, thus all derivations under the regression setting with analytical computation of KL terms, as well as the corresponding algorithms, generalize naturally into the classification settings. The code for training and inference is exactly the same. Discrete Denoising Diffusion Probabilistic Models (D3PMs) (Austin et al., 2021) fit conventional perception in classification tasks naturally by keeping the assumption of a categorical distribution. Therefore, the corresponding evaluation metrics like NLL can directly translate into such a framework — we believe that by adopting the discrete space assumption, a better NLL metric can be achieved. Meanwhile, it would require a lot more changes to be made from our framework for regression tasks, including the choice of the transition matrix, the incorporation of $\boldsymbol{x}$ into the diffusion processes, as well as the addition of the auxiliary loss into the objective function — all of the above are classification-task-specific settings, and cannot be adopted with the existing framework for regression tasks.

Besides the intention for consistency and generalizability across the two types of supervised learning tasks, we found that our current construction gives reasonable results to access model prediction confidence at the instance level — by directly using the prediction intervals obtained in the raw continuous space, *i.e.*, before adopting the softmax function for conversion to probability space, we obtain the sharp contrast in true label PIW between correct and incorrect predictions, and can already achieve high accuracy by merely predicting the label with the narrowest PIW for each instance. However, after converting the reconstructed labels to the probability space, the true label PIW contrast is reduced drastically, and the prediction accuracy by the narrowest PIW is similar to a random guess.

To recap, if achieving the best NLL and ECE for classification is the goal, then we think discrete diffusion models like Austin et al. (2021) could be excellent choices due to their use of the cross-entropy loss that is directed related to NLL and ECE; however, if the main goal is to access the prediction confidence at the instance level, the proposed CARD framework works well, and it would be interesting to make a head-to-head comparison with discrete diffusion-based classification models that yet need to be developed.

### A.2.3 Comparing CARD with Kendall and Gal (2017)

Kendall and Gal (2017) address BNN as an important class of methods for modeling uncertainty. CARD is similar to BNNs in providing stochastic outputs. However, BNNs deliver such stochasticity by modeling *epistemic* uncertainty, the uncertainty over network parameters $\boldsymbol{W}$ (by placing a prior distribution over $\boldsymbol{W}$) — this type of uncertainty is a *property of the model*. On the other hand, CARD does not model epistemic uncertainty, as it applies a deterministic deep neural network as its functional form; it is designed to model *aleatoric* uncertainty instead, which is a *property intrinsic to the data*. In Eq. 2 of Kendall and Gal (2017), such aleatoric uncertainty is captured by the last term as $\sigma^2$, which is a constant with respect to the network parameters $\theta$ for the variational distribution of model parameter $\boldsymbol{W}$, thus ignored during the optimization of $\theta$. The new method proposed in Kendall and Gal (2017) aims to model the aleatoric uncertainty by making $\sigma^2$ as part of the BNN output (Eq. 7); however, note that it still explicitly assumes $p(\boldsymbol{y} \mid \boldsymbol{x})$ to be a Gaussian distribution, as the objective function is the negative Gaussian log-likelihood, thus its effectiveness in capturing the actual aleatoric uncertainty depends on the validity of such parametric assumption for $p(\boldsymbol{y} \mid \boldsymbol{x})$.

### A.2.4 Comparing CARD with Score-Based Generative Classifiers

From a naming perspective, it might be easy to confuse CARD for classification as a type of *generative classifiers*, as it utilizes a generative model to conduct classification tasks. However, they are two different types of generative models, as generative classifiers model the conditional distribution $p(\boldsymbol{x} \,|\, \boldsymbol{y})$, while CARD models a different conditional distribution, *i.e.*, $p(\boldsymbol{y} \,|\, \boldsymbol{x})$. In fact, CARD shall be categorized as a type of discriminative classifier, by the definition in Zimmermann et al. (2021). Note that although both types of classifiers under image-based tasks would report NLL as one evaluation metric, they are also different, since the NLL for generative classifiers is evaluated in the space transformed from the logit space of $\boldsymbol{x}$, while the NLL for discriminative classifiers is computed in the space of $\boldsymbol{y}$ as the cross-entropy between the true label and the predicted probability.

### A.3 Classification on FashionMNIST Dataset

We perform classification on FashionMNIST dataset with CARD, and present the results in a similar fashion as Section 4.2.2. We first contextualize the performance of CARD through the accuracy of other BNNs with the LeNet CNN (LeCun et al., 1998) architecture in Table 6, where the metrics were first reported in Tomczak et al. (2021). For this dataset, our pre-trained classifier has the same LeNet architecture as the baselines, which achieves a test accuracy of $91.12\%$. CARD improves the mean test accuracy to $91.79\%$.

Table 6: Comparison of accuracy (in $\%$) on FashionMNIST dataset with other BNNs.

| Model | CMV-MF-VI | CM-MF-VI | CV-MF-VI | CM-MF-VI OPT | MF-VI | MAP | MC Dropout | MF-VI EB | $f_\phi$ (LeNet) | CARD |
|---|---|---|---|---|---|---|---|---|---|---|
| Accuracy | $91.10 \pm 0.22$ | $90.95 \pm 0.31$ | $88.53 \pm 0.13$ | $90.67 \pm 0.07$ | $87.04 \pm 0.28$ | $88.06 \pm 0.22$ | $87.99 \pm 0.17$ | $87.04 \pm 0.08$ | $91.12$ | $\mathbf{91.79 \pm 0.09}$ |

We then present Table 7, from which we can draw the same conclusions as Table 5 on the CIFAR-10 dataset. Note that we set $\alpha = 0.01$ for the paired two-sample $t$-test. When making the prediction for each instance merely by the class with the narrowest PIW, we obtain a test accuracy of $89.36\%$.

Table 7: PIW (multiplied by $100$) and $t$-test results for FashionMNIST classification task.

| Class | Accuracy | PIW | | Accuracy by $t$-test Status | |
|---|---|---|---|---|---|
| | | Correct | Incorrect | Rejected | Not-Rejected (Count) |
| All | 91.79% | 0.67 | 3.20 | 92.07% | 55.84% (77) |
| 1 | 88.10% | 0.96 | 3.40 | 88.45% | 61.54% (13) |
| 2 | 98.50% | 0.39 | 2.08 | 98.60% | 66.67% (3) |
| 3 | 87.70% | 0.84 | 3.42 | 88.00% | 50.00% (8) |
| 4 | 91.10% | 0.76 | 2.97 | 91.59% | 53.85% (13) |
| 5 | 87.90% | 0.89 | 2.91 | 88.40% | 33.33% (9) |
| 6 | 97.20% | 0.41 | 2.89 | 97.29% | 66.67% (3) |
| 7 | 74.80% | 1.37 | 3.26 | 74.90% | 70.00% (20) |
| 8 | 97.40% | 0.49 | 1.60 | 97.50% | 50.00% (2) |
| 9 | 98.40% | 0.34 | 1.93 | 98.50% | 0.00% (1) |
| 10 | 96.80% | 0.46 | 5.59 | 97.09% | 40.00% (5) |

### A.4 Test for Normality Assumption of Paired Two-Sample $t$-test

To assess the normality assumption of the paired two-sample $t$-test, we inspect the Q-Q plots of the differences in probability between the most and the second most predicted classes within each test instance. We include 16 instances in Figure 2 from CARD predictions on FashionMNIST dataset, each with 100 samples. We observe that in all plots, the points align closely with the $45$-degree line, indicating that the normality assumption is valid.

### A.5 Patch Accuracy vs Patch Uncertainty (PAvPU)

Besides the methods of assessing model confidence at the instance level introduced in this paper, we also consider an additional uncertainty evaluation metric in this section, *Patch Accuracy vs Patch*

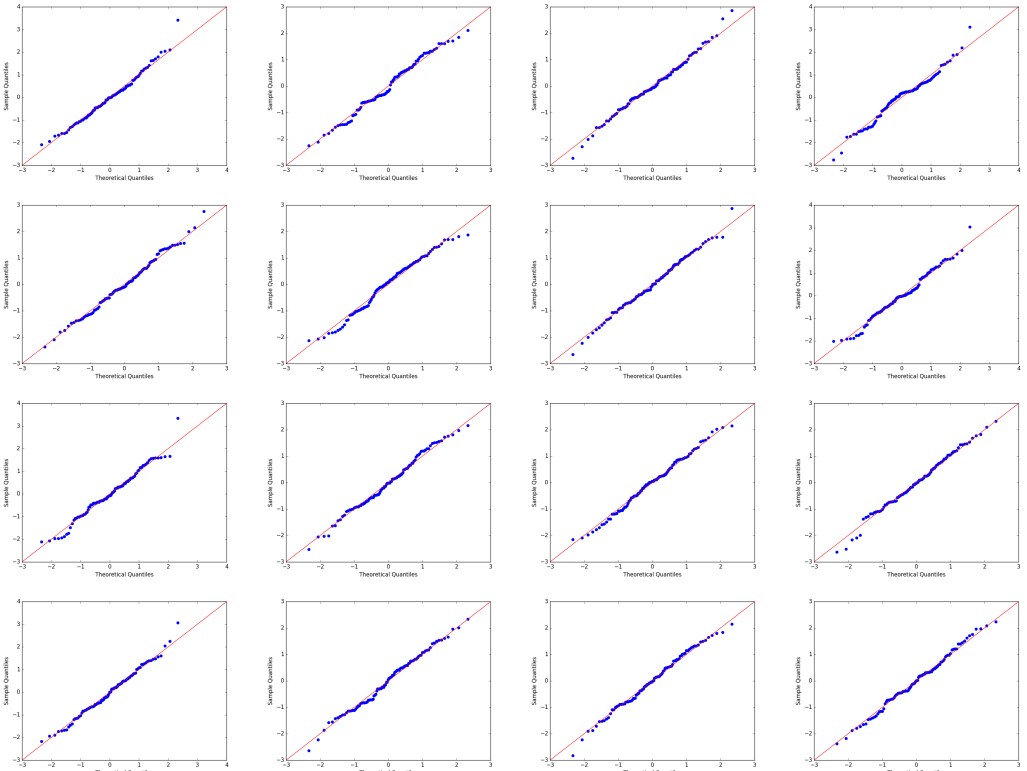

Figure 2: Q-Q plots for the differences in probability between the most and $2^{nd}$ most predicted class.

*Uncertainty* (PAvPU) (Mukhoti and Gal, 2018), which measures the proportion of predictions that are either correct while the model is confident about them, or incorrect while the model is ambiguous. Based on the $t$-test results, we can easily compute PAvPU, which is defined as

$$\text{PAvPU} \coloneqq \frac{n_{ac} + n_{iu}}{n_{ac} + n_{au} + n_{ic} + n_{iu}}, \tag{30}$$

where $n_{ac}, n_{au}, n_{ic}, n_{iu}$ represent the number of accurate (correct) predictions when the model is certain (confident) about them, accurate but uncertain, inaccurate but certain, as well as inaccurate and uncertain. We apply the $t$-test results as the proxy for the model's confidence level on each of its predictions. A higher value indicates that the model tends to be correct when being confident, and to make mistakes when being vague — a characteristic that we want our model to possess. On a related note, accuracy can be computed by replacing $n_{iu}$ with $n_{au}$ in Eq. (30). In the following section, we provide additional experimental results including this metric.

## A.6 Classification on Noisy MNIST dataset

To further demonstrate the effectiveness of the CARD model for classification, especially in expressing model confidence, we run additional experiments on Noisy MNIST dataset (adding a Gaussian noise with mean 0 and variance 1 to each pixel). Besides reporting PIW and $t$-test results similar to Table 5, we also compute PAvPU, and compare the results with baseline models in Fan et al. (2021), *i.e.*, MC Dropout (Gal and Ghahramani, 2016), Gaussian Dropout (Srivastava et al., 2014), Concrete Dropout (Gal et al., 2017), Bayes by Backprop (Blundell et al., 2015), and variants of Contextual Dropout (Fan et al., 2021).

Following the same experimental settings in Fan et al. (2021), we apply a batch size of 128 for model training, and adopt the same MLP architecture with two hidden layers of 300 and 100 hidden units, respectively, each followed by a ReLU non-linearity, as the network architecture for the pre-trained classifier $f_\phi(\boldsymbol{x})$. We simplify the diffusion network architecture (detailed in Section A.8) by reducing feature dimension from 2048 to 128 to match the model parameter scale for a fair comparison. We

pre-train $f_\phi(\boldsymbol{x})$ for 100 epochs, and obtain a test set accuracy of 85.50%. We train the diffusion model for 1000 epochs. Unlike on CIFAR-10 or FashionMNIST dataset, we do not apply data normalization on the noisy MNIST dataset, following the settings in Fan et al. (2021).

Similar to Table 5, we report the mean PIW among correct and incorrect predictions, and the mean accuracy among instances rejected and not-rejected by the paired two-sample $t$-test, in Table 8. Same as Fan et al. (2021), we set $\alpha = 0.05$. We report these metrics at the scope of the entire test set, and for each true class label along with their group accuracy.

Table 8: PIW (multiplied by 100) and $t$-test ($\alpha = 0.05$) results for Noisy MNIST classification task.

| Class | Accuracy | PIW | | Accuracy by $t$-test Status | |
|---|---|---|---|---|---|
| | | Correct | Incorrect | Rejected | Not-Rejected (Count) |
| All | 88.26% | 39.76 | 76.25 | 91.24% | 43.04% (618) |
| 1 | 94.69% | 15.93 | 68.67 | 96.63% | 35.48% (31) |
| 2 | 96.48% | 20.48 | 74.73 | 97.66% | 46.15% (26) |
| 3 | 88.47% | 29.92 | 74.98 | 91.25% | 44.26% (61) |
| 4 | 87.23% | 41.38 | 78.58 | 90.85% | 45.68% (81) |
| 5 | 85.34% | 48.87 | 78.61 | 87.65% | 53.73% (67) |
| 6 | 85.76% | 51.35 | 76.43 | 89.85% | 40.54% (74) |
| 7 | 92.28% | 22.92 | 74.04 | 94.49% | 28.13% (32) |
| 8 | 86.97% | 39.80 | 74.42 | 90.17% | 37.10% (62) |
| 9 | 80.90% | 62.48 | 77.73 | 84.25% | 45.88% (85) |
| 10 | 83.25% | 76.92 | 76.67 | 87.69% | 42.42% (99) |

From Table 8, we are able to draw similar conclusions as in Table 5: across the entire test set, mean PIW of the true class label among the correct predictions is much narrower than that of the incorrect predictions, implying that CARD is confident in its correct predictions, and tends to make mistakes when being vague. When comparing mean PIWs at true class level, we observe that a more accurate class is inclined to have a larger difference between correct and incorrect predictions. Meanwhile, similar to Table 5, the accuracy of test instances rejected by the $t$-test is much higher than that of the not-rejected ones, both across the entire test set and at the class level, while there are almost 10 times of not-rejected cases as those for CIFAR-10 task. This result could have a more significant practical impact when applying human-machine collaboration: we are able to identify more than 6% of the data with a mean accuracy of less than 50% — if we pass these cases to human agents, we would be able to remarkably improve the classification accuracy, while still enjoying the automation by the machine classifier in the vast majority of test instances.

Furthermore, we contextualize the performance of CARD by reporting accuracy, PAvPU and NLL, along with those from the baseline models mentioned at the beginning of Section A.6, in Table 9. The metrics of the other models are from Table 1 in Fan et al. (2021).

Table 9: PAvPU (in %) along with accuracy (in %) and NLL for Noisy MNIST classification.

| Method | Accuracy | PAvPU ($\alpha = 0.05$) | NLL |
|---|---|---|---|
| MC - Bernoulli | 86.36 | 85.63 | 1.72 |
| MC - Gaussian | 86.31 | 85.64 | 1.72 |
| Concrete | 86.52 | 86.77 | 1.68 |
| Bayes by Backprop | 86.55 | 87.13 | 2.30 |
| Contextual Gating | 86.20 | - | 1.81 |
| Contextual Gating + Dropout | 86.70 | 87.01 | 1.71 |
| Bernoulli Contextual | 87.43 | 87.81 | 1.41 |
| Gaussian Contextual | 87.35 | 87.72 | 1.43 |
| CARD (ours) | **88.26** | **89.12** | **0.39** |

We observe that CARD obtains an accuracy of 88.26% (improves from 85.50% by the pre-trained classifier $f_\phi$), and a PAvPU of 89.12%, both are the best among all models. This implies that our model is making not only accurate classifications, but also high-quality predictions in terms of model confidence. Lastly, we apply temperature scaling (Guo et al., 2017) to calibrate the predicted probability for the computation of NLL, where the temperature parameter is tuned with the training set, and again obtain the best metric among all models.

## A.7 Classification on Large-Scale Benchmark Datasets

In this section, we demonstrate the generalizability of CARD on classification tasks with our proposed framework of evaluating instance-level model prediction confidence, particularly on large-scale datasets, by reporting the experimental results on CIFAR-100 (100 classes, $10,000$ test instances), ImageNet-100 (100 classes, $5,000$ test instances), and ImageNet (1000 classes, $50,000$ test instances).

For the CIFAR-100 dataset, we pre-train the deterministic base classifier $f_\phi$ with a ResNet-18 architecture, which achieves a test accuracy of $71.37\%$. For the ImageNet-100 dataset, we apply a ResNet-50 architecture and use the parameters of a linear classifier fine-tuned with the self-supervised recipe (He et al., 2020; Zheng et al., 2021) for $f_\phi$, which achieves a test accuracy of $82.30\%$. For the ImageNet dataset, we adopt three different training paradigms for $f_\phi$, all using a ResNet-50 architecture, but obtain different test accuracies: the first one has its parameters learned with the self-supervised pipeline, *i.e.*, the feature encoder of the ResNet-50 network is firstly pre-trained in a self-supervised manner with the loss proposed by Zheng et al. (2021), and then by fixing the encoder the linear classifier of the network is further tuned with label supervision — this $f_\phi$ achieves a test accuracy of $73.87\%$; the second one reads the pre-trained weights provided by TorchVision, with the paradigm specified by Liang et al. (2022), which achieves a test accuracy of $76.13\%$; the third one also reads the pre-trained weights provided by TorchVision, with the paradigm specified by Pham and Le (2021), which achieves a test accuracy of $80.30\%$.

For each of the three large-scale benchmark datasets, we conduct 10 experimental runs, and report both majority-voted accuracy and PAvPU (with $\alpha = 0.01$ for the paired two-sample $t$-test) by CARD, along with the accuracy by the corresponding $f_\phi$, in Table 11. We observe that under all circumstances, CARD is able to improve the accuracy from the base classifier $f_\phi$.

For the CIFAR-100 dataset with CARD, we also contextualize the performance of CARD through the accuracy of other BNNs with ResNet-18 architecture in Table 10, where the metrics were reported in Tomczak et al. (2021).

Table 10: Comparison of accuracy (in $\%$) on CIFAR-100 dataset with other BNNs.

| Model | CMV-MF-VI | CM-MF-VI | CV-MF-VI | MF-VI | MC Dropout | MAP | $f_\phi$ (ResNet-18) | CARD |
|---|---|---|---|---|---|---|---|---|
| Accuracy | $60.59 \pm 0.39$ | $59.61 \pm 0.37$ | $46.22 \pm 0.54$ | $40.54 \pm 0.72$ | $54.49 \pm 0.36$ | $52.08 \pm 0.34$ | $71.37$ | $\mathbf{71.42 \pm 0.01}$ |

Regarding instance-level model prediction uncertainty assessment, we pick one run for each dataset, and report PIW among correct and incorrect predictions, at the entire test set level and at the true class level. Since each dataset has either 100 or 1000 total number of classes, we only report PIW for the class with the most and the least accurate predictions, as well as the mean accuracy given the $t$-test rejection status at the whole test set level. Furthermore, we record the test accuracy when making the prediction for each instance merely by the class with the narrowest PIW. We report these metrics for all three large-scale datasets in Table 12. For all experimental runs, we are able to draw conclusions consistent with those from the smaller-scale benchmark datasets (CIFAR-10 with Table 5, FashionMNIST with Table 7, and Noisy MNIST with Table 8): relative variability in label reconstruction captured by true label PIW is strongly related to prediction correctness, with sharper contrast between correct and incorrect predictions in a more accurate class; meanwhile, the accuracy of instances not rejected by the $t$-test is much lower than that of the rejected ones, indicating a great potential of improvement in prediction accuracy if combined with human inspection for the not-rejected cases.

## A.8 General Experiment Setup Details

In this section, we provide the experimental setup for the CARD model in both regression and classification tasks.

**Training**: As mentioned at the beginning of Section 4, we set the number of timesteps to 1000, and adopt a linear $\beta_t$ schedule same as Ho et al. (2020). We set the learning rate to $0.001$ for all tasks. We use the AMSGrad (Reddi et al., 2018) variant of the Adam optimizer (Kingma and Ba, 2015) for all regression tasks. We use the Adam optimizer for the classification tasks on all presented datasets,

Table 11: Majority-voted accuracy and PAvPU by CARD on CIFAR-100, ImageNet-100, and ImageNet, over 10 experimental runs. Pre-trained base classifier $f_\phi$ accuracy is also reported.

| Dataset | Acc. by $f_\phi$ | Accuracy | PAvPU |
|---|---|---|---|
| CIFAR-100 | 71.37% | $71.42 \pm 0.01\%$ | $71.48 \pm 0.03\%$ |
| ImageNet-100 | 82.30% | $82.35 \pm 0.03\%$ | $82.73 \pm 0.07\%$ |
| ImageNet | 73.87% | $74.28 \pm 0.01\%$ | $74.63 \pm 0.02\%$ |
| ImageNet | 76.13% | $76.20 \pm 0.00\%$ | $76.29 \pm 0.01\%$ |
| ImageNet | 80.30% | $80.35 \pm 0.01\%$ | $80.55 \pm 0.01\%$ |

Table 12: PIW (multiplied by 100), accuracy by predicting with the narrowest PIW, and accuracy by $t$-test rejection status, for the CIFAR-100, ImageNet-100, and ImageNet classification tasks, over a single experimental run. Due to the number of classes, we only report the PIW for all test instances, and within the most and least accurate classes. We applied multiple $f_\phi$ for the ImageNet dataset.

| Dataset | Accuracy | PIW Correct | PIW Incorrect | Acc. by PIW | Acc. by $t$-test Result Rejected | Not-Rejected (Count) |
|---|---|---|---|---|---|---|
| CIFAR-100 | | | | | | |
| overall | 71.42% | 0.59 | 3.91 | 60.53% | 71.56% | 35.90% (39) |
| most acc. | 95.00% | 0.16 | 1.92 | | | |
| least acc. | 44.00% | 5.09 | 5.84 | | | |
| ImageNet-100 | | | | | | |
| overall | 82.34% | 2.06 | 13.73 | 68.64% | 82.90% | 34.48% (58) |
| most acc. | 98.00% | 0.72 | 8.06 | | | |
| least acc. | 42.00% | 6.79 | 14.15 | | | |
| ImageNet ($f_\phi$ Accuracy 73.87%) | | | | | | |
| overall | 74.28% | 0.65 | 3.11 | 69.22% | 74.63% | 24.93% (349) |
| most acc. | 98.00% | 0.27 | 2.80 | | | |
| least acc. | 8.00% | 20.10 | 50.07 | | | |
| ImageNet ($f_\phi$ Accuracy 76.13%) | | | | | | |
| overall | 76.20% | 0.51 | 3.60 | 75.21% | 76.30% | 25.71% (105) |
| most acc. | 98.00% | 0.08 | 2.66 | | | |
| least acc. | 18.00% | 1.87 | 3.26 | | | |
| ImageNet ($f_\phi$ Accuracy 80.30%) | | | | | | |
| overall | 80.35% | 1.42 | 5.13 | 74.08% | 80.59% | 27.63% (228) |
| most acc. | 98.00% | 0.49 | 2.34 | | | |
| least acc. | 8.00% | 91.70 | 84.61 | | | |

and adopt cosine learning rate decay (Loshchilov and Hutter, 2017). We use exponentially weighted moving averages (Cox, 1961) on model parameters with a decay factor of 0.9999. We adopt antithetic sampling (Ren et al., 2019) to draw correlated timesteps during training. We set the number of epochs at 5000 by default for regression tasks, and 1000 by default for classification tasks, to sufficiently cover the timesteps with $T = 1000$. The batch size for each UCI regression task are reported in Table 15. We also apply a batch size of 256 for all toy regression tasks and classification tasks. For all UCI regression tasks, we follow the convention in Hernández-Lobato and Adams (2015) and standardize both the input features and the response variable to have zero mean and unit variance, and remove the standardization to compute the metrics. For all toy regression tasks, we do not standardize the input feature; we only standardize the response variable on the log-log cubic regression task. For classification on CIFAR-10 and FashionMNIST, we normalize the dataset with the mean and standard deviation of the training set.

**Network Architecture**: For the diffusion model, we adopt a simpler network architecture than that in previous work (Xiao et al., 2022; Zheng et al., 2022), by first changing the Transformer sinusoidal

position embedding to linear embedding for the timestep. As the network $\boldsymbol{\epsilon}_\theta\big(\boldsymbol{x}, \boldsymbol{y}_t, f_\phi(\boldsymbol{x}), t\big)$ has three other inputs besides the timestep $t$, we integrate them in different ways for regression and classification tasks:

- For regression, we first concatenate $\boldsymbol{x}$, $\boldsymbol{y}_t$, and $f_\phi(\boldsymbol{x})$, then send the resulting vector through three fully-connected layers, all with an output dimension of $128$. We perform Hadamard product between each of the output vector with the corresponding timestep embedding, followed by a Softplus non-linearity, before sending the resulting vector to the next fully-connected layer. Lastly, we apply a $4$-th fully-connected layer to map the vector to one with a dimension of $1$, as the output forward diffusion noise prediction. We summarize the architecture in Table 13 (a).

- For classification on CIFAR-10, we first apply an encoder on the flattened input image (originally $32 \times 32 \times 3$) to obtain a representation with $4096$ dimensions. The encoder consists of three fully-connected layers with an output dimension of $4096$. Meanwhile, we concatenate $\boldsymbol{y}_t$ and $f_\phi(\boldsymbol{x})$, and apply a fully-connected layer to obtain an output vector of $4096$ dimensions. We perform a Hadamard product between such vector and a timestep embedding to obtain a response embedding conditioned on the timestep. We then perform Hadamard product between image embedding and response embedding to integrate these variables, and send the resulting vector through two more fully-connected layers with $4096$ output dimensions, each would first followed by a Hadamard product with a timestep embedding, and lastly a fully-connected layer with an output dimension of $1$ as the noise prediction. Note that all fully-connected layers are also followed by a batch normalization layer and a Softplus non-linearity, except the output layer. We summarize the architecture in Table 13 (b).

Table 13: CARD $\boldsymbol{\epsilon}_\theta$ network architecture. We denote concatenation as $\oplus$, Hadamard product as $\odot$, and Softplus non-linearity as $\sigma$. We also denote a fully-connected layer as $g$, and a hidden layer output as $l$, with subscripts to differentiate them.

(a) Regression network architecture.

| input: $\boldsymbol{x}, \boldsymbol{y}_t, f_\phi(\boldsymbol{x}), t$ |
| :---: |
| $l_1 = \sigma\Big(g_{1,a}\big(\boldsymbol{x} \oplus \boldsymbol{y}_t \oplus f_\phi(\boldsymbol{x})\big) \odot g_{1,b}(t)\Big)$ |
| $l_2 = \sigma\big(g_{2,a}(l_1) \odot g_{2,b}(t)\big)$ |
| $l_3 = \sigma\big(g_{3,a}(l_2) \odot g_{3,b}(t)\big)$ |
| output: $g_4(l_3)$ |

(b) Classification network architecture.

| input: $\boldsymbol{x}, \boldsymbol{y}_t, f_\phi(\boldsymbol{x}), t$ |
| :---: |
| $l_{1,x} = \sigma\Big(\mathrm{BN}\big(g_{1,x}(\boldsymbol{x})\big)\Big)$ |
| $l_{2,x} = \sigma\Big(\mathrm{BN}\big(g_{2,x}(\boldsymbol{x})\big)\Big)$ |
| $l_{3,x} = \mathrm{BN}\big(g_{1,x}(\boldsymbol{x})\big)$ |
| $l_{1,y} = \sigma\Big(\mathrm{BN}\Big(g_{1,y}\big(\boldsymbol{y}_t \oplus f_\phi(\boldsymbol{x})\big) \odot g_{1,b}(t)\Big)\Big)$ |
| $l_1 = l_{3,x} \odot l_{1,y}$ |
| $l_2 = \sigma\Big(\mathrm{BN}\big(g_{2,a}(l_1) \odot g_{2,b}(t)\big)\Big)$ |
| $l_3 = \sigma\Big(\mathrm{BN}\big(g_{3,a}(l_2) \odot g_{3,b}(t)\big)\Big)$ |
| output: $g_4(l_3)$ |

For the pre-trained model $f_\phi(\boldsymbol{x})$, we adjust the functional form and training scheme based on the task. For regression, we adopt a feed-forward neural network with two hidden layers, each with $100$ and $50$ hidden units, respectively. We apply a Leaky ReLU non-linearity with a $0.01$ negative slope after each hidden layer. In practice, we find that when the dataset size is small (most of the UCI tasks have less than $10,000$ data points, and several around or less than $1000$), a deep neural network $f_\phi(\boldsymbol{x})$ is prone to overfitting. We thus set the default number of epochs to $1000$, and adopt early stopping with a patience of $50$ epochs, *i.e.*, we terminate the training if there is no improvement in the validation set MSE for $50$ consecutive epochs. We split the original training set with $60\%/40\%$ ratio and apply early stopping to find the optimal number of epochs, then train $f_\phi(\boldsymbol{x})$ on the full training set. For classification, we apply a pre-trained ResNet-18 network for CIFAR-10 dataset. We only train the model with $10$ epochs, as more would lead to overfitting. We apply the Adam optimizer for $f_\phi(\boldsymbol{x})$ in all tasks, and we only pre-train the model and freeze it during the training of the diffusion model, instead of fine-tuning it.

### A.9 UCI Baseline Model Experiment Setup Details and Dataset Information

In this section, we first provide the experimental setup for the UCI regression baseline models (PBP, MC Dropout, Deep Ensembles, and GCDS), including learning rate, batch size, network architecture, number of epochs, etc. We applied the GitHub repo (Joachims, 2021) to run BNN models, and implemented our own version of GCDS (Zhou et al., 2021) since the code has not been published. We note that GCDS is related to a concurrent work of Yang et al. (2022), who share a comparable idea but use it in a different application: regularizing the learning of an implicit policy in offline reinforcement learning.

Overall, we apply the same learning rate of $0.001$ for all models on all datasets, except for PBP on the Boston dataset, which is $0.1$. We also apply the Adam optimizer for all experiments. We follow the convention in Hernández-Lobato and Adams (2015) and standardize both the input features and response variable for training, and remove the standardization for evaluation.

For batch size, we adjust on a case-by-case basis, taking the running time and dataset size into consideration. We provide the batch size in Table 15. For network architecture, we use ReLU non-linearities for all 3 BNNs, and Leaky ReLU with a $0.01$ negative slope for GCDS; we choose the number of hidden layers with the number of hidden units per layer from the following 3 options: a) 1 hidden layer with 50 hidden units; b) 1 hidden layer with 100 hidden units; c) 2 hidden layers with 100 and 50 hidden units, respectively. We show the choice of hidden layer architecture in Table 16. For the number of training epochs, we also vary on a case-by-case basis: PBP and Deep Ensembles both applied 40 epochs, but we observed in many experiments that the model has not converged. We show the number of epochs in Table 17. Lastly, datasets Boston, Energy, and Naval all contain one or more categorical variables, thus we ran the experiments both with and without conducting one-hot encoding on the data. We found that except the Naval dataset with PBP, all other cases had worse metrics when one-hot encoding was applied.

We summarize the dataset information in terms of their size and number of features in Table 14.

Table 14: Dataset size ($N$ observations, $P$ features) of UCI regression tasks.

| Dataset | Boston | Concrete | Energy | Kin8nm | Naval | Power | Protein | Wine | Yacht | Year |
|---------|--------|----------|--------|--------|-------|-------|---------|------|-------|------|
| $(N, P)$ | $(506, 13)$ | $(1030, 8)$ | $(768, 8)$ | $(8192, 8)$ | $(11, 934, 16)$ | $(9568, 4)$ | $(45, 730, 9)$ | $(1599, 11)$ | $(308, 6)$ | $(515, 345, 90)$ |

Table 15: Batch size settings of UCI regression tasks across different models.

|  | PBP | MC Dropout | Deep Ensembles | GCDS | CARD (ours) |
|---------|-----|------------|----------------|------|-------------|
| Boston | 32 | 32 | 32 | 32 | 32 |
| Concrete | 32 | 32 | 32 | 32 | 32 |
| Energy | 32 | 32 | 32 | 32 | 32 |
| Kin8nm | 64 | 32 | 64 | 64 | 64 |
| Naval | 64 | 32 | 64 | 64 | 64 |
| Power | 64 | 64 | 64 | 64 | 64 |
| Protein | 100 | 256 | 100 | 256 | 256 |
| Wine | 32 | 32 | 32 | 32 | 32 |
| Yacht | 32 | 32 | 32 | 32 | 32 |
| Year | 256 | 256 | 100 | 256 | 256 |

We reiterate that we re-ran the experiments with the baseline BNN models to compute the new metric QICE along with the conventional metrics RMSE and NLL. We carefully tuned the hyperparameters to obtain results better than or comparable with the ones reported in the original papers.

### A.10 UCI Regression Tasks PICP across All Methods

In this section, we report PICP for all methods in Table 18 from the same runs with the corresponding metrics in Tables 1, 2, and 3.

Table 16: Network hidden layer architecture for UCI regression tasks across different models.

|          | PBP | MC Dropout | Deep Ensembles | GCDS |
|----------|-----|------------|----------------|------|
| Boston   | a   | c          | a              | c    |
| Concrete | a   | c          | c              | c    |
| Energy   | a   | b          | a              | c    |
| Kin8nm   | a   | c          | a              | a    |
| Naval    | a   | c          | a              | a    |
| Power    | a   | b          | a              | a    |
| Protein  | b   | c          | b              | c    |
| Wine     | a   | c          | a              | c    |
| Yacht    | a   | a          | a              | a    |
| Year     | b   | c          | b              | b    |

Table 17: Number of training epochs of UCI regression tasks across different models.

|          | PBP | MC Dropout | Deep Ensembles | GCDS |
|----------|-----|------------|----------------|------|
| Boston   | 100 | 1000       | 100            | 500  |
| Concrete | 100 | 500        | 40             | 500  |
| Energy   | 100 | 500        | 100            | 500  |
| Kin8nm   | 100 | 500        | 100            | 500  |
| Naval    | 100 | 500        | 100            | 500  |
| Power    | 100 | 500        | 100            | 500  |
| Protein  | 100 | 500        | 100            | 500  |
| Wine     | 100 | 500        | 100            | 500  |
| Yacht    | 100 | 500        | 100            | 500  |
| Year     | 100 | 100        | 100            | 500  |

## A.11  Ablation Study on Choice of Prior — UCI Boston Dataset

In this section, we conduct ablation study for two model variants with different prior distribution settings on the UCI Boston dataset, under various settings of the number of timesteps $T$ with adjusted $\beta_t$ linear schedule (to make sure that $\sqrt{\bar{\alpha}_1}$ is close to 1 and $\sqrt{\bar{\alpha}_T}$ is close to 0). We report the evaluation metrics in Table 19, where we compare the original CARD setting with $\mathcal{N}(f_\phi(\boldsymbol{x}), \boldsymbol{I})$ as the prior distribution at timestep $T$, to the alternative setting with $\mathcal{N}(\boldsymbol{0}, \boldsymbol{I})$ as the prior distribution. We observe that as $T$ decreases, RMSE and NLL do not deteriorate for $\mathcal{N}(f_\phi(\boldsymbol{x}), \boldsymbol{I})$ prior (CARD setting), but those from $\mathcal{N}(\boldsymbol{0}, \boldsymbol{I})$ prior become worse. The metrics that measure distributional fitting, QICE and PICP, gets worse under the $\mathcal{N}(f_\phi(\boldsymbol{x}), \boldsymbol{I})$ prior setting as well, but such deterioration is not as much as $\mathcal{N}(\boldsymbol{0}, \boldsymbol{I})$ prior. The results indicate that our setting of an informative prior $\mathcal{N}(f_\phi(\boldsymbol{x}), \boldsymbol{I})$ contributes to the regression performance of CARD. Furthermore, the setting of the total number of timesteps $T$ does not affect the mean estimation for $\mathcal{N}(\boldsymbol{0}, \boldsymbol{I})$ prior, but would noticeably impact the distributional fitting (*i.e.*, the recovery of aleatoric uncertainty).

## A.12  Ablation Study on Diffusion Network Parameterization — CIFAR-10 Dataset

In this section, we study the impact of different $\epsilon_\theta$ network parameterization forms on the CIFAR-10 dataset, through model performance in terms of accuracy and PAvPU, as well as training efficiency at the first 100 epochs. We compare four model variants, each with a different prior and $\epsilon_\theta$ network parameterization combination, in Table 20, by reporting accuracy and PAvPU on the test set over 10 runs.

We observe that given the same prior distribution setting, both metrics do not differ much by whether or not we include $f_\phi(\boldsymbol{x})$ as the input of the $\epsilon_\theta$ network. Meanwhile, both model variants (V1, V2) with a prior of $\mathcal{N}(f_\phi(\boldsymbol{x}), \boldsymbol{I})$ outperform the other two variants (V3, V4) of $\mathcal{N}(\boldsymbol{0}, \boldsymbol{I})$ prior, suggesting the application of an informative prior would benefit the performance. Furthermore, the variant (V4) with $f_\phi(\boldsymbol{x})$ as neither the prior mean nor $\epsilon_\theta$ input has the worst performance, indicating the inclusion of a pre-trained classifier can improve the model performance in both accuracy and uncertainty estimation.

Furthermore, the choice of $\mathcal{N}(f_\phi(\boldsymbol{x}), \boldsymbol{I})$ prior also helps with training efficiency: we observe in Figure 3 that the model performance improved faster for $\mathcal{N}(f_\phi(\boldsymbol{x}), \boldsymbol{I})$ prior at the beginning of

Table 18: PICP (in %) of UCI regression tasks.

| Dataset | | | $\| \text{PICP} - 95 \| \downarrow$ | | |
| | PBP | MC Dropout | Deep Ensembles | GCDS | CARD (ours) |
|---|---|---|---|---|---|
| Boston | $91.27 \pm 4.82$ | $\mathbf{96.08 \pm 2.70}$ | $88.73 \pm 5.68$ | $31.37 \pm 6.79$ | $93.24 \pm 3.59$ |
| Concrete | $92.28 \pm 2.87$ | $\mathbf{97.52 \pm 2.43}$ | $90.34 \pm 3.64$ | $39.85 \pm 4.53$ | $90.24 \pm 3.45$ |
| Energy | $93.18 \pm 3.12$ | $99.03 \pm 1.08$ | $\mathbf{96.49 \pm 1.97}$ | $63.57 \pm 10.26$ | $98.70 \pm 1.30$ |
| Kin8nm[1] | $\mathbf{95.06 \pm 0.77}$ | $95.37 \pm 2.24$ | $96.53 \pm 0.67$ | $59.06 \pm 5.31$ | $93.68 \pm 0.79$ |
| Naval[2] | $93.52 \pm 4.40$ | $100.00 \pm 0.00$ | $99.78 \pm 0.28$ | $83.71 \pm 14.87$ | $\mathbf{95.35 \pm 0.60}$ |
| Power | $95.75 \pm 0.69$ | $96.28 \pm 0.76$ | $95.91 \pm 0.71$ | $89.13 \pm 1.14$ | $\mathbf{94.87 \pm 0.65}$ |
| Protein | $\mathbf{94.79 \pm 0.13}$ | $96.46 \pm 0.77$ | $96.08 \pm 0.28$ | $85.24 \pm 0.86$ | $95.38 \pm 0.16$ |
| Wine | $92.72 \pm 1.80$ | $91.41 \pm 2.66$ | $91.06 \pm 1.64$ | $86.37 \pm 2.33$ | $\mathbf{93.88 \pm 2.10}$ |
| Yacht | $\mathbf{96.94 \pm 2.60}$ | $100.00 \pm 0.00$ | $98.87 \pm 1.54$ | $83.23 \pm 4.28$ | $99.84 \pm 0.70$ |
| Year | $93.04 \pm$ NA | $\mathbf{94.61 \pm}$ NA | $95.44 \pm$ NA | $87.08 \pm$ NA | $93.35 \pm$ NA |
| # best | $\mathbf{3}$ | $\mathbf{3}$ | 1 | 0 | $\mathbf{3}$ |

Table 19: Ablation study on 2 prior distribution settings on UCI Boston dataset with different $T$.

| T | $\beta_t$ schedule $(\beta_1, \beta_T)$ | Prior | RMSE | NLL | QICE | PICP |
|---|---|---|---|---|---|---|
| 1000 | (0.0001, 0.02) | $\mathcal{N}(f_\phi(\boldsymbol{x}), \boldsymbol{I})$ | $\mathbf{2.61 \pm 0.63}$ | $2.65 \pm 0.12$ | $\mathbf{3.45 \pm 0.83}$ | $93.24 \pm 3.59$ |
| | | $\mathcal{N}(\mathbf{0}, \boldsymbol{I})$ | $2.71 \pm 0.69$ | $\mathbf{2.37 \pm 0.12}$ | $3.53 \pm 0.99$ | $\mathbf{93.53 \pm 3.34}$ |
| 500 | (0.0001, 0.04) | $\mathcal{N}(f_\phi(\boldsymbol{x}), \boldsymbol{I})$ | $\mathbf{2.63 \pm 0.72}$ | $\mathbf{2.33 \pm 0.13}$ | $3.94 \pm 1.05$ | $\mathbf{93.14 \pm 3.19}$ |
| | | $\mathcal{N}(\mathbf{0}, \boldsymbol{I})$ | $2.70 \pm 0.68$ | $2.34 \pm 0.12$ | $\mathbf{3.48 \pm 0.76}$ | $91.76 \pm 3.75$ |
| 100 | (0.001, 0.175) | $\mathcal{N}(f_\phi(\boldsymbol{x}), \boldsymbol{I})$ | $\mathbf{2.65 \pm 0.67}$ | $\mathbf{2.30 \pm 0.18}$ | $4.09 \pm 1.13$ | $\mathbf{88.82 \pm 5.15}$ |
| | | $\mathcal{N}(\mathbf{0}, \boldsymbol{I})$ | $2.69 \pm 0.66$ | $2.32 \pm 0.21$ | $4.19 \pm 1.12$ | $85.20 \pm 6.34$ |
| 50 | (0.001, 0.35) | $\mathcal{N}(f_\phi(\boldsymbol{x}), \boldsymbol{I})$ | $\mathbf{2.61 \pm 0.71}$ | $\mathbf{2.31 \pm 0.25}$ | $\mathbf{5.06 \pm 1.46}$ | $\mathbf{81.96 \pm 6.31}$ |
| | | $\mathcal{N}(\mathbf{0}, \boldsymbol{I})$ | $2.76 \pm 0.66$ | $2.57 \pm 0.39$ | $5.38 \pm 1.55$ | $76.18 \pm 7.13$ |
| 10 | (0.01, 0.95) | $\mathcal{N}(f_\phi(\boldsymbol{x}), \boldsymbol{I})$ | $\mathbf{2.63 \pm 0.58}$ | $\mathbf{2.56 \pm 0.44}$ | $\mathbf{5.34 \pm 1.24}$ | $\mathbf{77.65 \pm 7.00}$ |
| | | $\mathcal{N}(\mathbf{0}, \boldsymbol{I})$ | $2.80 \pm 0.75$ | $2.98 \pm 0.85$ | $5.52 \pm 1.20$ | $75.39 \pm 7.58$ |

training, by measuring the accuracy on the test set with 1 sample every 10 epochs for the first 100 epochs during training, and plotting the metric (as the mean across all runs) against the number of epochs. Due to the measurement similarity, we omit V2 and V4 and only plot the metrics from V1 (for $\mathcal{N}(f_\phi(\boldsymbol{x}), \boldsymbol{I})$ prior) and V3 (for $\mathcal{N}(\mathbf{0}, \boldsymbol{I})$ prior). We observe that after only 20 epochs, the accuracy of 1 sample by CARD is already close to 90%, while the variant with a $\mathcal{N}(\mathbf{0}, \boldsymbol{I})$ prior is only around 75%, suggesting the advantage in training efficiency with an informative prior.

### A.13 Regression Toy Example Details

The 8 toy examples are summarized by Table 21. For each task, we create the dataset by sampling $10,240$ data points from the data generating function, and randomly split them into training and test sets with an $80\%/20\%$ ratio. For all uni-modal cases as well as the full circle task, the $\boldsymbol{x}$ variable is sampled from a uniform distribution. The noise variable $\epsilon$ is sampled from a Gaussian distribution. The dataset of the inverse sinusoidal task is created by simply swapping $\boldsymbol{x}$ and $\boldsymbol{y}$ variable of the sinusoidal task (so that we have multi-modality when the new $\boldsymbol{x}$ is roughly between $0.25$ and $0.75$), thus the name of the task.

To quantitatively evaluate the performance of CARD, we generate $1000$ $\boldsymbol{y}$ samples for each $\boldsymbol{x}$ in the test set, and compute the corresponding metrics. We conduct such a procedure over 10 runs, each applying a different random seed to generate the dataset, and report the mean and standard deviation over all runs for each metric. For all tasks regardless of the form of $p(\boldsymbol{y} \mid \boldsymbol{x})$, we compute PICP and QICE. For tasks with uni-modal $p(\boldsymbol{y} \mid \boldsymbol{x})$ distributions, we summarize the 1000 samples for each test $\boldsymbol{x}$ by computing their mean, as an unbiased estimator to $\mathbb{E}(\boldsymbol{y} \mid \boldsymbol{x})$, and compute the root mean squared error (RMSE) between the estimated and true conditional means. For all tasks, we obtain a mean PICP very close to the optimal $95\%$, and most of the tasks have a mean QICE value far less than

---

[1] The data generating function of this task was originally proposed in Bishop (1994).

[2] Swap $\boldsymbol{x}$ and the generated $\boldsymbol{y}$ from the sinusoidal regression task.

[3] We set the coordinates of 8 modes at $(\sqrt{2}, 0)$, $(-\sqrt{2}, 0)$, $(0, \sqrt{2})$, $(0, -\sqrt{2})$, $(1, 1)$, $(1, -1)$, $(-1, 1)$, $(-1, -1)$, and add a noise sample to both coordinates of a mode to generate one instance.

Table 20:  4 Ablation study on 4 model variants on CIFAR-10 dataset.

| Variant | Prior | $f_\phi(\boldsymbol{x})$ included as $\epsilon_\theta$ input | Accuracy | PAvPU |
|---------|-------|---------------------------------------------------------------|----------|-------|
| V1 | $\mathcal{N}(f_\phi(\boldsymbol{x}), \boldsymbol{I})$ | True | $90.93 \pm 0.02$ | $\mathbf{91.11 \pm 0.04}$ |
| V2 | $\mathcal{N}(f_\phi(\boldsymbol{x}), \boldsymbol{I})$ | False | $\mathbf{90.94 \pm 0.02}$ | $91.08 \pm 0.03$ |
| V3 | $\mathcal{N}(\boldsymbol{0}, \boldsymbol{I})$ | True | $90.88 \pm 0.03$ | $91.06 \pm 0.03$ |
| V4 | $\mathcal{N}(\boldsymbol{0}, \boldsymbol{I})$ | False | $90.82 \pm 0.02$ | $91.02 \pm 0.03$ |

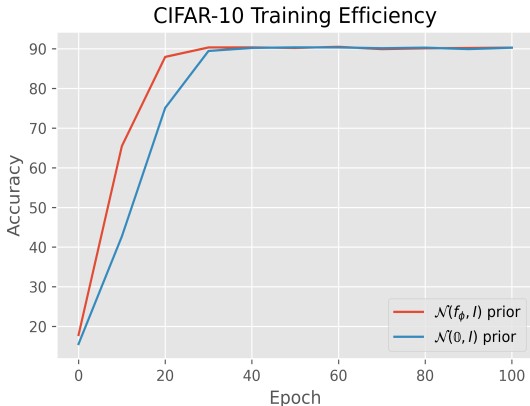

Figure 3: Performance from two prior settings on CIFAR-10 test set with 1 sample.

0.01 except log-log cubic regression, which also has a mean RMSE noticeably larger by an order of magnitude among cases with uni-modal conditional distributions. Note that the $\boldsymbol{y}$ samples here have a much wider range: as $\boldsymbol{x}$ increases from 0 to 10, $\boldsymbol{y}$ increases from 0 to over 1200, resulting in a much more difficult task. Therefore, the metrics reported here can be viewed with relativity, and combined with the qualitative conclusions from Figure 1. The metrics of all tasks, including RMSE, QICE, and PICP, are recorded in Table 22. Note that QICE has been converted to a percentage scale as we report two significant figures for all metrics.

The results in Table 22 implicitly suggest that our proposed metric QICE is reasonable: when RMSE is low (way below 1) and PICP is close to $95\%$, implying that CARD is performing well in terms of both mean estimation and distributional matching, QICE is also low (far less than $1\%$); for the most difficult task, log-log cubic regression, as RMSE is above 5 and PICP deviates relatively most from $95\%$ (but not much), QICE also has the largest value (slightly above $1\%$).

### A.14   The Evolution of Samples through the Diffusion Process

We present the evolution of both $q$ and $p$ distribution samples through the forward and reverse diffusion process, respectively. We first visualize the behaviors of these samples from the training on linear regression tasks in Figure 4, where we pick timesteps with an interval of 200 steps including the $1^{st}$ and the last timestep, namely $t = 1, 200, 400, 600, 800, T$. The $p$ samples presented are from near the end of training. We observe that $\epsilon_\theta$ has been trained to match the $q$ samples at different timesteps well, including the variance. Furthermore, note that the true variance from the data generating function is set to 4, while the prior $p(\boldsymbol{y}_T \mid \boldsymbol{x})$ has a variance of 1. We can observe the gradual increase of variance in the reverse direction. This example helps to illustrate that when $f_\phi(\boldsymbol{x})$ can already estimate the mean accurately, it makes the task for the diffusion model easier: in this case to solely focus on recovering the aleatoric uncertainty.

Similarly, we present the samples from $q$ and $p$ distribution during training for the full circle regression task in Figure 5. Besides observing the matching in samples at all selected timesteps, we emphasize CARD's ability to model multi-modality at various intensities. As $t$ increases, we observe the samples gradually evolve from a uni-modal distribution into a multi-modal one, and the diffusion model is able to capture such progress. To quantify such match, we plot the quantile coverage ratios for samples at $t = 0$ from one run, along with the optimal coverage ratio (0.1, for 10 bins), in Figure 6.

Table 21: Regression toy examples.

| Regression Task | Data Generating Function | $x$ | $\epsilon$ |
|---|---|---|---|
| Linear | $y = 2x + 3 + \epsilon$ | $U(-5, 5)$ | $\mathcal{N}(0, 2^2)$ |
| Quadratic | $y = 3x^2 + 2x + 1 + \epsilon$ | $U(-5, 5)$ | $\mathcal{N}(0, 2^2)$ |
| Log-Log Linear | $y = \exp\big(\log(x) + \epsilon\big)$ | $U(0, 10)$ | $\mathcal{N}(0, 0.15^2)$ |
| Log-Log Cubic | $y = \exp\big(3\log(x) + \epsilon\big)$ | $U(0, 10)$ | $\mathcal{N}(0, 0.15^2)$ |
| Sinusoidal[1] | $y = x + 0.3\sin(2\pi x) + \epsilon$ | $U(0, 1)$ | $\mathcal{N}(0, 0.08^2)$ |
| Inverse Sinusoidal[2] | swap $x$ and $y$ from Sinusoidal | — | — |
| 8 Gaussians[3] | 8 modes | — | $\mathcal{N}(0, 0.1^2)$ |
| Full Circle | $y = (10 + \epsilon)\big(\cos(2\pi x) + \sin(2\pi x)\big)$ | $U(0, 1)$ | $\mathcal{N}(0, 0.5^2)$ |

Table 22: Regression toy example RMSE, QICE (in %), and PICP (in %).

| Regression Task | RMSE ↓ | QICE ↓ | PICP |
|---|---|---|---|
| Linear | $0.07 \pm 0.02$ | $0.54 \pm 0.14$ | $95.29 \pm 0.53$ |
| Quadratic | $0.21 \pm 0.03$ | $0.55 \pm 0.12$ | $95.12 \pm 0.55$ |
| Log-Log Linear | $0.07 \pm 0.01$ | $0.55 \pm 0.15$ | $95.17 \pm 0.62$ |
| Log-Log Cubic | $5.85 \pm 1.38$ | $1.31 \pm 0.26$ | $96.08 \pm 0.62$ |
| Sinusoidal | $0.01 \pm 0.00$ | $0.48 \pm 0.11$ | $94.81 \pm 0.54$ |
| Inverse Sinusoidal | — | $0.71 \pm 0.18$ | $95.89 \pm 0.52$ |
| 8 Gaussians | — | $0.66 \pm 0.19$ | $95.92 \pm 0.46$ |
| Full Circle | — | $0.60 \pm 0.05$ | $95.52 \pm 0.42$ |

Note that we obtain a QICE of $0.62$ in this run. We observe that CARD samples cover the true data with a ratio close to the optimal across all bins; The $5^{th}$ bin has relatively the most deviation from the optimal ratio (which is understandable as the full circle dataset has a bi-modal distribution across most $x$, with no data points in the center portion), which is compensated by the $2^{nd}$, $4^{th}$ and last bin with coverage slightly above the optimal ratio.

We continue with a plot of samples during test time from the UCI Boston dataset. In Figure 7, we plot the generated samples from $p$ (in blue) along with $q$ samples (in red) at various $t$. The $x$-axis represents the count of samples, instead of the actual $x$ since the true covariate space is high-dimensional. Note that we generate 1000 samples given each $x$. While still observing the good mix between $q$ and $p$ samples for $t = 200, \ldots, T$ from $2^{nd}$ plot to the right, we observe that for $t = 1$, all samples for each $x$ forms a vertical region that covers the corresponding $y_1$ (which shall be very close to $y_0$ due to the linear $\beta_t$ schedule) from $q$ distribution at various positions (*i.e.*, middle, upper half, lower half, near top, near bottom). We observe that the samples are representative of the true conditionals $p(y \mid x = x)$ for each $x$.

## A.15  The Development of Model Performance through the Reverse Diffusion Process

Following the demonstration of behavior change in samples with respect to timestep $t$ during both the training and test time, we present the change in model evaluation metrics on the test set as a function of $t$. We again use the UCI Boston dataset, and compute all evaluation metrics, including RMSE, NLL, QICE, and PICP, at all timesteps from $t = T$ to $t = 0$. Since the standard procedure for running the task contains 20 different splits on the dataset, we compute these metrics at all timesteps for all splits, and take the mean across all splits at each timestep $t$. We combine these plots in Figure 8. For RMSE, we observe that the performance at $t = T$ is already quite good, due to the setting of $f_\phi(x)$ as the prior mean; as the reverse process continues, the metric gradually improves by decreasing. Both NLL and QICE behave in a similar fashion: as $t$ decreases, the metric steadily improves. For PICP, note that it crosses the optimal value of $0.95$ coverage ratio around the end of the reverse process; however, it did not deviate much from such value. This plot demonstrates the successive improvements across all metrics during the reverse diffusion process, starting from an already decent place (in terms of RMSE and NLL) due to the application of an informative prior.

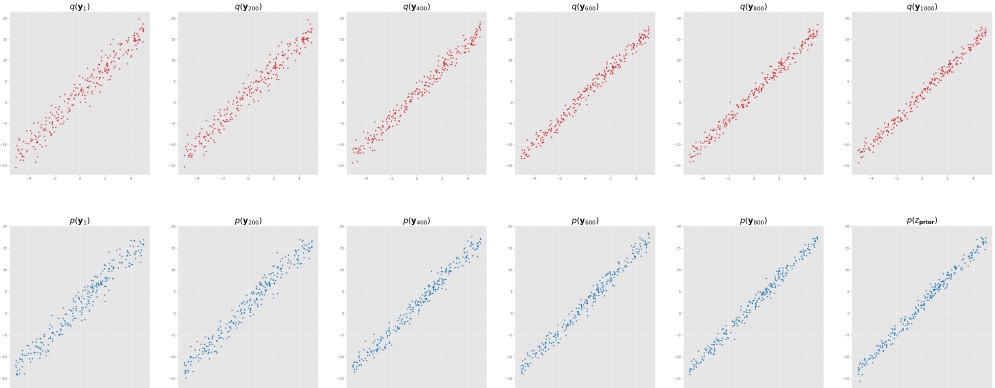

Figure 4: $q$ and $p$ distribution samples for linear regression task during training. (**Top**) left to right: $q\big(\boldsymbol{y}_t \,|\, \boldsymbol{y}_0, f_\phi(\boldsymbol{x})\big)$ for $t = 1, 200, \ldots, T$; (**Bottom**) right to left: $p_\theta(\boldsymbol{y}_{t-1} \,|\, \boldsymbol{y}_t, \boldsymbol{x})$ for $t = T, \ldots, 200, 1$.

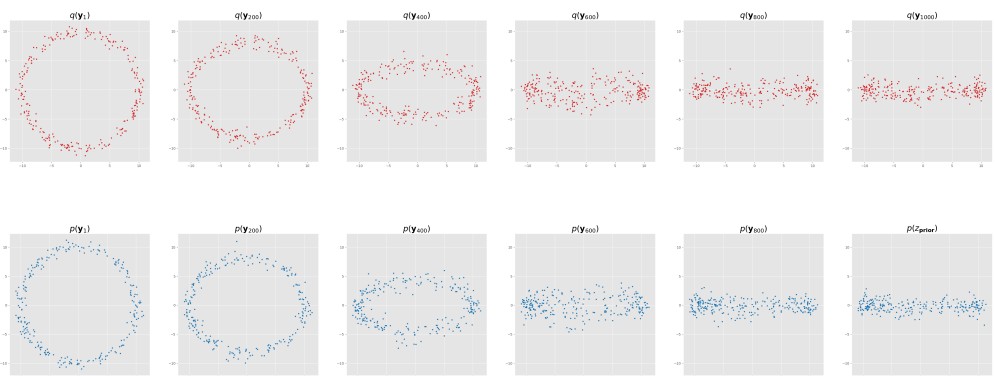

Figure 5: $q$ (top) and $p$ (bottom) distribution samples for full circle regression task during training.

## A.16 Improving the Granularity of ECE from Subgroup to Instance Level

In this section, we first present the definition of ECE. This material is from Guo et al. (2017), and we include it here for completeness. We then provide our analysis, specifically about its granularity to measure prediction confidence by the model, which motivates us to introduce an alternative way to measure model prediction confidence at a finer granularity (*i.e.*, at instance level) in our paper.

ECE is defined as:

$$\text{ECE} := \mathbb{E}_{\hat{P}}\big[\big|\mathbb{P}\big(\hat{Y} = Y \,|\, \hat{P} = p\big) - p\big|\big], \tag{31}$$

where $Y$ and $\hat{Y}$ are true and predicted class labels, respectively; $\hat{P}$ is the predicted probability associated with $\hat{Y}$. A perfect calibration is defined as:

$$\mathbb{P}\big(\hat{Y} = Y \,|\, \hat{P} = p\big) = p, \ \forall p \in [0, 1]. \tag{32}$$

However, since the predicted probability $\hat{Y}$ is continuous in $[0, 1]$, we cannot compute ECE with finite instances, thus we approximate it by first **dividing the probability space into $M$ bins with equal width**, then compute the confidence and accuracy within each bin. Each test instance is placed into one specific bin by the predicted probability value associated with the true class label. For the $m$-th bin $B_m$, we have accuracy (proportion of correct predictions)

$$\text{acc}(B_m) = \frac{1}{|B_m|} \sum_{i \in B_m} \mathbb{1}(\hat{y}_i = y_i) \tag{33}$$

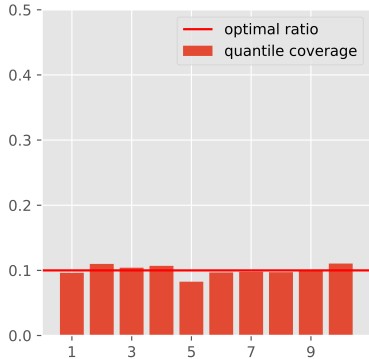

Figure 6: Sample coverage ratio by bins for full circle regression task (QICE 0.62).

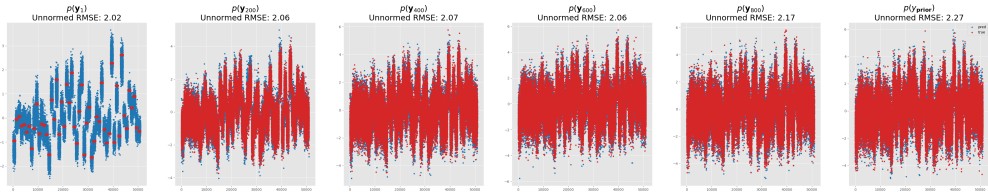

Figure 7: $q$ (red) and $p$ (blue, 1000 samples per $\boldsymbol{x}$) samples from UCI Boston test set.

and confidence (mean of predicted probabilities)

$$\text{conf}(B_m) = \frac{1}{|B_m|} \sum_{i \in B_m} \hat{p}_i, \tag{34}$$

where $\hat{y}_i$ and $\hat{p}_i$ are the predicted label and its associated probability value, and $y_i$ is the true label, for instance $i$. We thus have the empirical version of ECE as:

$$\text{ECE} \coloneqq \sum_{m=1}^{M} \frac{|B_m|}{n} \big| \text{acc}(B_m) - \text{conf}(B_m) \big|, \tag{35}$$

where $|B_m|$ and $n$ are the cardinality of the $m$-th bin and the total number of instances, respectively.

To summarize, although we are interested in measuring the miscalibration through the difference between $p(y_i = \hat{y}_i \,|\, \boldsymbol{x}_i)$ and $\hat{p}_i$, we are only able to compute such miscalibration *at the granularity of subgroup level* — usually with the number of subgroups $M$ set to 10 or less in practice. In other words, we cannot make a statement with the existing classification framework like the following: given this new test instance, we predict the class label to be (some class), but we are very sure our prediction is correct. This observation motivates us to introduce an alternative way of measuring model confidence, *at the granularity of the instance level*.

### A.17   Insights on How CARD Accurately Recovers the Conditional Distributions

Although a theoretical justification is not within the scope of our paper, in this section we plan to briefly talk about what makes CARD stands out as a great candidate to model $p(\boldsymbol{y} \,|\, \boldsymbol{x}, \mathcal{D})$, through the capability to model implicit distributions by diffusion models in general: the theory of stochastic processes by Feller (1949) suggests that:

a) With a large enough number of timesteps $T$, $q(\boldsymbol{x}_T \,|\, \boldsymbol{x}_0)$ would converge to a stationary distribution $p(\boldsymbol{x}_T)$ regardless of the distribution at timestep $t = 0$, $q(\boldsymbol{x}_0)$. In other words, we are able to go from any distribution $q(\boldsymbol{x}_0)$ to a stationary distribution $p(\boldsymbol{x}_T)$ by choice.

b) Meanwhile, with a large enough $T$ and small enough noise schedule $\{\beta_t\}_{t=1:T}$, the product in $q(\boldsymbol{x}_{t-1} \,|\, \boldsymbol{x}_t) \propto q(\boldsymbol{x}_t \,|\, \boldsymbol{x}_{t-1})q(\boldsymbol{x}_{t-1})$ would be dominated by $q(\boldsymbol{x}_t \,|\, \boldsymbol{x}_{t-1})$, thus both forward and reverse diffusion processes would share the same functional form. Although $q(\boldsymbol{x}_{t-1} \,|\, \boldsymbol{x}_t)$ cannot be

easily estimated, if we are able to learn a function $p_\theta(\boldsymbol{x}_{t-1} \mid \boldsymbol{x}_t)$ that approximates $q(\boldsymbol{x}_{t-1} \mid \boldsymbol{x}_t)$ well, we are able to reverse the direction mentioned in a) and go from $p(\boldsymbol{x}_T)$ to any $q(\boldsymbol{x}_0)$.

For real world datasets like the ones in Dua and Graff (2017), the relationship between the covariates and response variable can be quite complicated. The class of diffusion models in general, including CARD, places no restriction on the parametric form for $q(\boldsymbol{x}_0)$ (*i.e.*, it fits the data as it is), therefore it is suitable to the situations where a flexible distribution assumption is needed, instead of just the ones where an explicit distribution assumption is valid (*e.g.*, Gaussian for BNNs).

### A.18 Insights on Why CARD Outperforms Other BNN Approaches

We observe the following two limitations of the class of BNN approaches:

a) It places an explicit distributional form assumption to $p(\boldsymbol{y} \mid \boldsymbol{x}, \boldsymbol{W})$, where $\boldsymbol{W}$ denotes the model parameter: for regression, it assumes an additive Gaussian noise model for $p(\boldsymbol{y} \mid \boldsymbol{x}, \boldsymbol{W})$ (Eq. 2 in Kendall and Gal (2017)) — we have addressed the limitation of such additive-noise assumption in the first two paragraphs of Section 1.

b) BNNs do not directly fit the actual posterior $p(\boldsymbol{W} \mid \boldsymbol{X}, \boldsymbol{Y})$, which is required for evaluating the marginal $p(\boldsymbol{Y} \mid \boldsymbol{X})$, due to its intractability, but rather fit an approximated distribution $q(\boldsymbol{W})$ that minimizes its KL divergence to the actual posterior. Such distribution $q(\boldsymbol{W})$ has a simple distributional assumption (usually Gaussian), which places another layer of restrictions for BNNs to model their predictive distribution $p(\boldsymbol{y} \mid \boldsymbol{x}) = \int p(\boldsymbol{y} \mid \boldsymbol{x}, \boldsymbol{W}) p(\boldsymbol{W} \mid \boldsymbol{X}, \boldsymbol{Y}) d\boldsymbol{W}$. CARD is able to dodge both limitations through the modeling of implicit distributions, as elaborated in Section A.17.

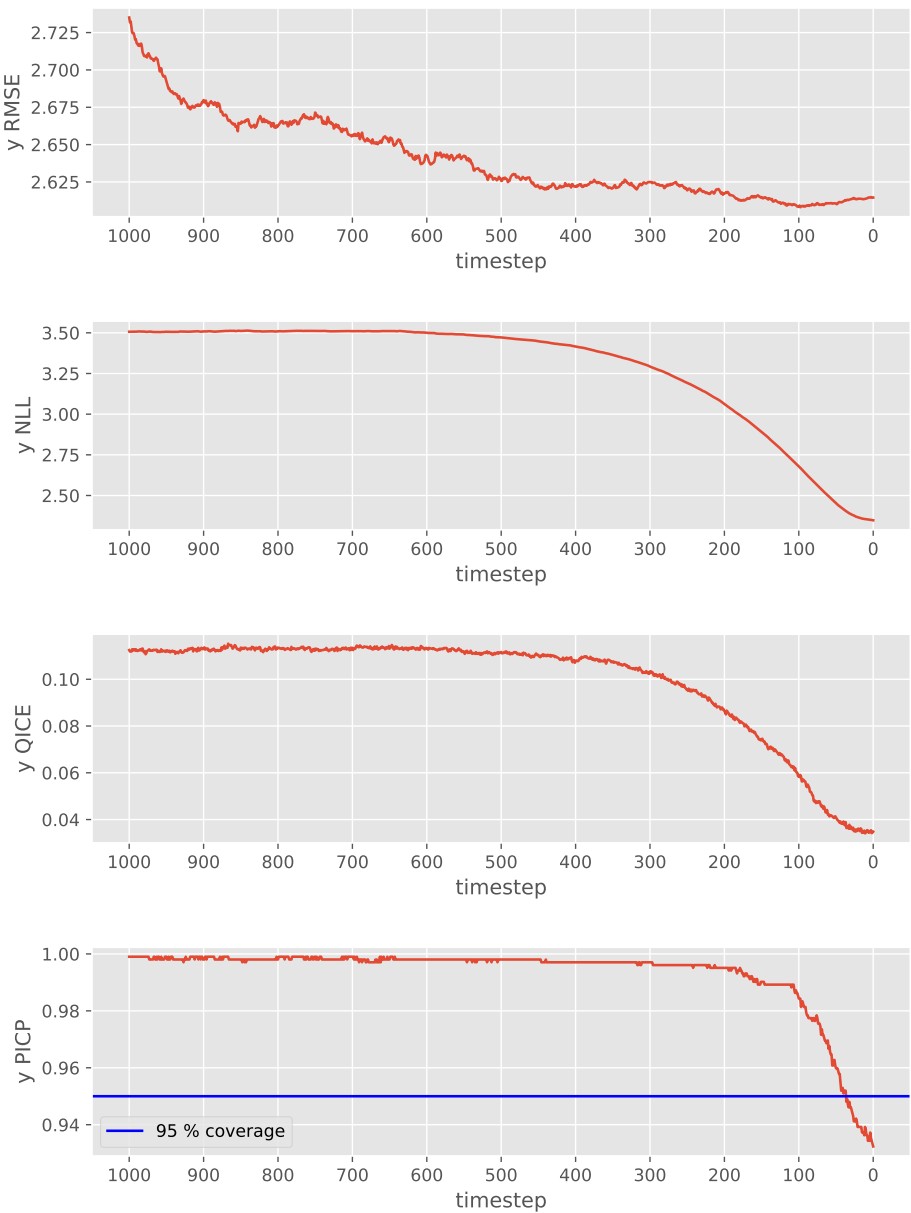

Figure 8: Change in regression evaluation metrics on UCI Boston dataset. The value at each timestep is the mean across all 20 splits.

# B  Broader Impact and Limitations

We believe that our model has a practical impact in various industrial settings, where supervised learning methods have been increasingly applied to facilitate the decision-making process. For regression tasks, *e.g.*, understanding the relationship between drug dosage and biographical features of a patient in the medical domain, and evaluating player performance given various on-court measurements, where the actual distribution of the real-world data is complicated and unknown, we are able to compute summary statistics with a minimal amount of assumptions, as we do not assume the parametric form of the conditionals $p(\boldsymbol{y} \,|\, \boldsymbol{x})$. The analytical results from our framework thus have the potential to reach a broader audience, who in the past might be hindered by the sheer amount of jargon due to the lack of statistical training. For classification tasks, as mentioned in both of the experiments, we could easily adopt a human-machine collaboration framework: since our model is capable of conveying the prediction confidence along with the prediction itself, we could pass the cases where the model is less assertive to humans for further evaluation. This trait is especially valuable for classification tasks with exceptionally imbalanced data, *e.g.*, fraud detection, and ad click-through rate prediction, where the volume of one class could be orders of magnitude more than the other. False negative errors for these applications are usually quite expensive, and simply adjusting the classification threshold would often put too many positive predictions for human agent evaluation. As demonstrated in our experiments, CARD is capable of providing uncertain cases with a very reasonable ratio, keeping the workload for human agents at a sensible level.

Meanwhile, since CARD is capable of modeling multi-modality, we are concerned that it could be devised for malicious purposes, like revealing personalized information of the patient through reverse engineering the prediction results: as an extension to the medical example of breast cancer given in the introduction section, one could tell with high confidence the gender of the patient based on the predicted mode. For research purposes, in this work, we only consider Gaussian diffusion and the reverse denoising in CARD, while there could be much more options when given different data. For example, in classification, we can optimize the classification likelihood with cross-entropy instead of using simple MSE loss, and directly perform diffusion in discrete spaces, as in Austin et al. (2021). We only take a pre-trained model $f_\phi$ as a deterministic neural network, while there could be more possibilities like combining BNN methods with CARD. Moreover, the computation efficiency of CARD may also be further investigated if the dataset size becomes larger. We encourage researchers in our community to further study those potential safety concerns and approaches for improvements in order to develop more mature supervised learning tools.

# C  Computational Resources

Our models are trained and evaluated with a single Nvidia GeForce RTX 3090 GPU. We use PyTorch (Paszke et al., 2019) 1.10.0 as the deep learning framework. CARD trains between 100 and 200 steps per second at the batch size specified in Table 15 for regression tasks, and at 44 steps per second at batch size 256 for classification on the CIFAR-10 dataset. The sampling for a batch of 250 test instances, each with 100 samples, takes around 1.05 seconds at most for the classification task on FashionMNIST. The computation time could vary if we apply different network architectures for regression and classification tasks, and the architecture details are provided in Appendix A.8.

Table 23: Model size and computation complexity of CARD and deterministic neural networks (architecture same as $f_\phi$) on different datasets. Throughputs measures the number of samples calculated per second. For UCI tasks, we measure the parameter size and computation complexity on the subset with the biggest data dimension.

| Task | Regression | | | | Classification | | | | | |
|---|---|---|---|---|---|---|---|---|---|---|
| Dataset | Toy | | UCI | | FMNIST | | CIFAR-10/100 | | ImageNet-100/1K | |
| model | #Params | Throughputs | #Params | Throughputs | #Params | Throughputs | #Params | Throughputs | #Params | Throuputs |
| CARD | 5.24e-2M | 25334.72 | 6.55e-2M | 244146.47 | 6.41e-2M | 1033.24 | 16.52M | 244.21 | 32.76M | 137.13 |
| DNN | 5.22e-2M | 25378.50 | 6.51e-2M | 250555.79 | 6.02e-2M | 1057.95 | 11.17M | 249.56 | 25.62M | 147.56 |

We summarize the model parameter size and measure the throughputs, *i.e.*, how many data samples can be proceeded per second, as computation complexity measure in Table 23. As our model consists of both a diffusion model $\epsilon_\theta$ in the response prediction and a pre-trained prior mean model $f_\phi$, we compute them respectively.