# OpenReview forum: "CARD: Classification and Regression Diffusion Models"
_NeurIPS.cc/2022/Conference — NeurIPS 2022 Accept_

### Official Review · Reviewer_8bC3 · 2022-07-09

**Rating:** 5
**Confidence:** 3
**Soundness:** 2 fair
**Presentation:** 2 fair
**Contribution:** 3 good

**Summary:**

This work applies diffusion models to discriminative task. The diffusion model is used to model the uncertainty of $y$. The author designs a special forward process to incorporate a pretrained discriminative model, which is the main distinction from prior diffusion models. Experiments valid that the CARD can model the uncertainty of the prediction.

**Questions:**



**Limitations:**



**Strengths And Weaknesses:**

Strength:
1. It is an interesting idea to apply diffusion models to discriminative tasks, due to its strong capacity to model the uncertainty.

2. The incorporation of a pretrained discriminative model seems a nice design.

Weakness:
1. The main difference between prior conditional diffusion models is that the transition in the forward process is related to the condition, i.e., use $q(y_t|y_{t-1}, x)$ instead of $q(y_t|y_{t-1})$. What will happen if we use $q(y_t|y_{t-1})$? Perhaps the author can consider $q(y_t|y_{t-1})$ as a baseline and make some comparison.

2. In Section 3.2.2, I didn't find a comparison of different methods on the classification uncertainty. While the author presents some uncertainty result of CARD, it is important to compare it with other methods. For example, a tranditional classifer can also capture the uncertainty, since it outputs the probability values of each class. These probabilities can be used for uncertainty evaluation. What is the uncertainty evaluation performance of CARD when comparing with the traditional classifier? I suggest the author run experiments in Table 6 for some other methods, and make a comparison.

3. The datasets used in this experiment are simple. How will CARD perform when scaling up to high resolution inputs? For example, CIFAR10 and ImageNet.


Typos:

1. In Eq.(1), $p_\theta(x, y_0)$ should be $p_\theta(y_0|x)$, and $q(y_{1:T}|x)$ should be $q(y_{1:T}|y_0, x)$.

2. The ELBO in Eq.(2) is also related to $y_0$, thus should be denoted as $L_{ELBO}(x, y_0)$ instead of $L_{ELBO}(x)$.

3. Eq.(6) is related to $x$, thus should be denoted as $p(y_T|x)$.

4. Eq.(9) is related to $y_0$, thus should be denoted as $q(y_{t-1}|y_t, y_0, f_\phi(x))$.

5. In L3 in Algorithm 1, $q(y_0)$ should be $q(y_0|x)$.

6. The $\epsilon_\theta$ between L6 and L7 in Algorithm 1 missed $f_\phi(x)$ as its input.

---

> ### Author Response · Authors · 2022-08-02
> **addressing questions and comments by Reviewer 8bC3 (Part. 2)**
>
> > Weakness 2: Comparison of different methods on the classification uncertainty
>
> Response 2: CARD is different from the traditional classifiers as although the latter would output a probability vector to capture label uncertainty, such probability prediction is in the form of a point estimate (i.e., no uncertainty on the probability vector itself). This is the main distinction between CARD and traditional classifiers, since given a covariate input $x$, CARD would generate its predicted probability vector in a stochastic fashion, thus one reverse diffusion process would output a probability-vector prediction different from another one, enabling us to construct an interval of probability predictions for each class label. With such intervals, we are able to continue measuring the model prediction confidence in terms of interval width and $t$-test results. For traditional classifiers with probability predictions that are point estimates, we are not able to construct such intervals.
>
> -------
>
> > Weakness 3: The datasets used in this experiment are simple. How will CARD perform when scaling up to high resolution inputs?
>
> Response 3: We have addressed this issue in our latest revision and highlighted the results of ImageNet (with resolution 224x224) in the common response. Please check the detailed results in our revision.
>
> ----------
>
> > Typos
>
> Response: Thank you for going through this section carefully. We really appreciate your pointing these typos out, and have made the corresponding modifications in our revised paper.

---

> ### Author Response · Authors · 2022-08-02
> **addressing questions and comments by Reviewer 8bC3 (Part. 1)**
>
> Dear Reviewer 8bC3,
>
> We appreciate your comments and questions -- please see below for our responses.
>
> > Weakness 1: The main difference between prior conditional diffusion models is that the transition in the forward process is related to the condition, i.e., use q(yt|yt−1,x) instead of q(yt|yt−1). What will happen if we use q(yt|yt−1)? Perhaps the author can consider q(yt|yt−1) as a baseline and make some comparison.
>
> Response 1: We appreciate you for asking this question, as we did actually start building CARD with a $q(y_t|y_{t-1})$ forward process setting, and had achieved comparable  results on toy regression datasets but worse results on other more complex settings. We added $x$ to the forward process and constructed the prior at timestep T to be $\mathcal{N}(f_{\phi}(x), I)$ instead of $\mathcal{N}(0, I)$ with the intuition s.t. a good $f_{\phi}(x)$ could very likely be closer to the true $y$ than $0$, it could make the sampling process more efficient, i.e., fewer number of timesteps T for good performance, faster convergence; and making the task for the conditional diffusion model easier, as now it just needs to capture the residual between true $y$ and $f_{\phi}(x)$. This is a summary of our findings:
>
> * On toy regression tasks, we didn’t observe much noticeable difference between adopting $\mathcal{N}(f_{\phi}(x), I)$ vs. $\mathcal{N}(0, I)$ as two types of priors, whether we add $f_{\phi}(x)$ as part of the $\epsilon_{\theta}$ network input, when we compared evaluation metrics at various T settings (with corresponding adjusted $\beta$ schedules) and at various number of epochs.
>
> * On UCI regression tasks, we observe that across various T settings, $\mathcal{N}(f_{\phi}(x), I)$ prior with $f_{\phi}(x)$ added as $\epsilon_{\theta}$ input obtains better metrics when comparing to $\mathcal{N}(0, I)$ prior without $f_{\phi}(x)$ added as $\epsilon_{\theta}$ input, and the performance does not deteriorate as much when T decreases.
>
> * On CIFAR-10 classification task, we compare 4 variants, with both prior settings combined with two $\epsilon_{\theta}$ input settings. We observed that for each prior settings, whether $f_{\phi}(x)$ is added as part of the $\epsilon_{\theta}$ input does not lead to significantly different performance; however, the two variants with $\mathcal{N}(f_{\phi}(x), I)$ prior outperformed the two with $\mathcal{N}(0, I)$ prior in terms of accuracy and PAvPU while the training also converges much faster at the initial epochs.
>
> To summarize, as the task at hand becomes more challenging – more complicated distributions for regression tasks, and higher dimensional response variables (number of classes instead of $1$) for classification tasks – the perks of applying $\mathcal{N}(f_{\phi}(x), I)$ (along with adding $f_{\phi}(x)$ as part of the $\epsilon_{\theta}$ input), including model performance in terms of evaluation metrics, and training efficiency, become more apparent.
>
> We report the ablation study on UCI Boston regression task with various T (Appendix A.10), as well as the comparison in terms of evaluation metrics and training efficiency for CIFAR-10 classification task (Appendix A.11).

---

### Official Review · Reviewer_VuVw · 2022-07-11

**Rating:** 5
**Confidence:** 4
**Soundness:** 3 good
**Presentation:** 2 fair
**Contribution:** 2 fair

**Summary:**

The paper presents a new perspective on the supervised learning of predicting a continuous or categorical response variable y given its covariates x.
Specifically, it proposes classification and regression diffusion (CARD) models, which combine a denoising diffusion-based conditional generative model and a pre-trained conditional mean estimator, to accurately predict the distribution of y given x.  The paper compares CARD with Bayesian neural network-based approaches on both toy examples and real-world datasets.

**Questions:**

The assumption behind
p(y_T ) = N (fφ (x), I ) (6)
need clarification. Please explain whether this applies to real-world datasets, e.g. ImageNet.  "where fφ(x) is pre-knowledge of the relation between x and y0, e.g., pre-trained with D to approximate E[y|x], or 0 if we assume the relation is unknown.". Does this mean fφ(x) can be computed by a DNN, e.g. ResNet50 for classification?

How does CARD compare with methods in "What Uncertainties Do We Need in Bayesian Deep Learning for Computer Vision?" NIPS 2017, ( Eqn. 2 for regression uncertainty estimation)?

Please discuss related work on generative classifiers, e.g. Score-Based Generative Classifiers (https://arxiv.org/abs/2110.00473).

If the issues can be adequately addressed in the revision, I would be happy to increase my rating from 4 to 5.

=== 8/7/2022 after rebuttal

I appreciate the authors' effort to address my comments. It clarified a few questions. I am happy to increase my rating to 5.

**Limitations:**

The limitations are not adequately stated. It only mentions that computation efficiency of CARD may also be further investigated if the dataset size becomes larger. While the preliminary empirical results look promising, in-depth experiments on typical real-world datasets, e.g. ImageNet needs to be studied. The authors need to connect with the literature of generative classifiers. Furthermore, there is a lack of insights or theoretical justification on the performance of CARD over discriminative models.

**Strengths And Weaknesses:**

Strengths

The paper reformulates the classical P(y | x): given the ground-truth response variable y0 and its covariates x, and assuming a sequence of intermediate uncertain prediction y1:T made by the diffusion model. It constructs εθ (x, yt , fφ (x), t), which is a function approximator parameterized by a deep neural network that predicts the forward diffusion noise ε sampled for yt. The training and inference procedure can be carried out in a standard DDPM manner.

The paper presents derivation for forward process posteriors and derivation for forward process sampling distribution with arbitrary time steps.

CARD is compared with MC Dropout, Gaussian Dropout, Concrete Dropout, Bayes by Backprop, and variants of Contextual Dropout on toy datasets, UCI regression tasks and Fashion MNIST classification tasks,

Weaknesses

The paper does not provide insights or theoretic justification on the reason that CARD can help accurately recovery p(y | x, D), the predictive distribution of y given x after observing data D.

The paper also does not offer insights on why CARD outperforms other Bayesian NN approaches.

The datasets used are very limited. The authors should consider real-world datasets such as ImageNet.

The paper should put their work in the context of related work better. This would require a related work section which is missing. For example, generative classifiers are not discussed.

---

> ### Author Response · Authors · 2022-08-02
> **addressing questions and comments by Reviewer VuVw (Part. 1)**
>
> Dear Reviewer VuVw,
>
> Thank you very much for your comments and questions. Please see below for our responses and answers.
>
> > Weaknesses 1: insights or theoretic justification for CARD
>
> Response 1: We appreciate your pointing it out, and plan to address through the capability to model implicit distributions by diffusion models in general: the theory of stochastic processes (W Feller, 1949) suggests that:
>
> a) With a large enough number of timesteps $T$, $q(x_T|x_0)$ would converge to a stationary distribution $p(x_T)$ regardless of the distribution at $t=0$, $q(x_0)$. In other words, we are able to go from any distribution $q(x_0)$ to a stationary distribution $p(x_T)$ by choice. (And in this paper, we choose $p(x_T)$ to be $\mathcal{N}(f_{\phi}(x), I)$).
>
> b) Meanwhile, with a large enough $T$ and small enough noise schedule $\beta_t$, the product in $q(x_{t-1}|x_t)\propto q(x_t|x_{t-1})q(x_{t-1}))$ would be dominated by $q(x_t|x_{t-1})$, thus both forward and reverse diffusion processes would share the same functional form. Although $q(x_{t-1}|x_t)$ cannot be easily estimated, if we are able to learn a function $p_{\theta}(x_{t-1}|x_t)$ that approximates $q(x_{t-1}|x_t)$ well, we are able to reverse the direction mentioned in 1) and go from $p(x_T)$ to any $q(x_0)$.
> For real world datasets like the UCI ones, the relationship between covariates and response variable can be quite complicated. The class of diffusion models in general, including CARD, places no restriction on the parametric form for $q(x_0)$ (i.e., it fits the data as it is), therefore it is capable of modeling more flexible distributions in general, instead of just the ones where some explicit distribution assumptions are valid (e.g., Gaussian for BNNs).
>
> As in this paper we focus more on demonstrating the performance by CARD specifically on supervised learning tasks, we leave the above discussion out. We are open to your suggestion as whether including such discussion is helpful for justifying the performance by CARD.
>
> ------
>
> > Weakness 2: The paper also does not offer insights on why CARD outperforms other Bayesian NN approaches.
>
> Response 2: We observe the following two limitations of the class of BNN approaches: 1) It places an explicit distributional form assumption to p(y|x,W), where W denotes the model parameter: for regression, it assumes an additive Gaussian noise model for  $p(y|x, \beta)$ (Eqn. 2 in Kendall and Gal (2017) as you mentioned in a later question) – we’ve addressed the limitation of such additive-noise assumption in the first two paragraphs of the Introduction section; 2) BNNs do not directly fit the actual posterior $p(W|X,Y)$ (required for evaluating the marginal p(Y|X)) due to its intractability, but rather fit an approximated distribution q(W) that minimizes its KL divergence to the actual posterior. Such distribution q(W) has a simple distributional assumption (usually Gaussian), which places another layer of restriction for BNNs to model their predictive distribution $p(y|x) = \int p(y|x, W)p(W|X,Y) dW$.
> CARD is able to dodge both limitations through the modeling of implicit distributions, as elaborated from our response to the previous point.

---

> > ### Author Response · Authors · 2022-08-02
> > **addressing questions and comments by Reviewer VuVw (Part. 2)**
> >
> > > Weakness 3: The datasets used are very limited.
> >
> > Response 3: Please see our common response and latest revision, where we provide results on ImageNet-100 and ImageNet-1K.
> >
> > ------
> >
> > > Weakness 4: Related work section.
> >
> > Response 4: We’ve added the Related Work session, currently placed in Appendix A.2 -- we've positioned generative classifiers as another class of methods that utilizes generative models to perform classification tasks.
> >
> > -------
> >
> > > Q1: The assumption behind p(y_T ) = N (fφ (x), I ) (6) need clarification.
> >
> > A1: Exactly – in the context of classification, $f_{\phi}(x)$ would represent a probability prediction for class label. The functional form can be chosen as one sees fit; and a deterministic deep neural network is a preferred choice by us, as when properly trained it can already obtain a satisfying accuracy. For our experiments in CIFAR-10 (and FashionMNIST, whose results are now placed in Appendix A.3), we apply a pre-trained ResNet18 network; for Noisy MNIST dataset, we apply a DNN with the same DNN architecture.
> >
> > --------
> >
> > > Q2: How does CARD compare with methods in "What Uncertainties Do We Need in Bayesian Deep Learning for Computer Vision?" NIPS 2017, ( Eqn. 2 for regression uncertainty estimation)?
> >
> > A2: The above-mentioned paper addresses Bayesian neural network (BNN) as an important class of methods for modeling uncertainty. CARD is related to BNNs in providing stochastic output. However, BNNs deliver such stochasticity by modeling *epistemic* uncertainty, the uncertainty over network parameters $W$ (by placing a prior distribution over $W$) – this type of uncertainty is a **property of the model**. On the other hand, CARD does not model epistemic uncertainty, as it applies a deterministic deep neural network as its functional form; it is designed to model *aleatoric* uncertainty instead, which is a **property intrinsic to the data**. In Eqn. 2 of the mentioned paper, such aleatoric uncertainty is captured by the last term as $\sigma^2$, which is a constant w.r.t. the network parameters $\theta$ for the variational distribution of model parameter $W$, thus ignored during the optimization of $\theta$. The new method proposed in this paper aims to model the aleatoric uncertainty by making $\sigma^2$ as part of the BNN output (Eqn. 7); however, note that it still explicitly assumes $p(y|x)$ to be a Gaussian distribution, as the objective function is the negative Gaussian log-likelihood, thus its effectiveness in capturing the actual aleatoric uncertainty depends on the validity of such parametric assumption for $p(y|x)$.
> >
> > ---------
> >
> > > Q3: Please discuss related work on generative classifiers, e.g. Score-Based Generative Classifiers (https://arxiv.org/abs/2110.00473).
> >
> > A3: From the naming perspective, it might be easy to confuse CARD for classification as a type of generative classifier, as it utilizes a generative model to conduct classification tasks. However, they are two different types of generative models, as generative classifiers model the conditional distribution $p(x|y)$, while CARD models a different conditional distribution, i.e., $p(y|x)$. In fact, CARD shall be categorized as a type of discriminative classifier, by the definition in the “Score-Based Generative Classifiers” paper. Note that although both types of classifiers under image-based tasks would report NLL as one evaluation metric, they are also different, since the NLL for generative classifiers is evaluated in the space transformed from the logit space of $x$, while the NLL for discriminative classifiers is computed in the space of $y$, as the cross-entropy between true labels and predicted probabilities.

---

> > > ### Comment · Reviewer_VuVw · 2022-08-07
> > > **Thank you**
> > >
> > > I appreciate the authors' effort to address my comments. It clarified a few questions. I am happy to increase my rating to 5. I would encourage the authors to address major limitations (e.g. insights or theoretic justification for CARD, experimental results on standard datasets such as ImageNet) in the final paper or future work.

---

### Official Review · Reviewer_RrLR · 2022-07-11

**Rating:** 6
**Confidence:** 3
**Soundness:** 3 good
**Presentation:** 3 good
**Contribution:** 3 good

**Summary:**

Diffusion models provide state of the art results in image and video synthesis and this paper extends them to uncertainty quantification for regression and classification problems. The proposed approach extends denoising diffusion probabilistic models to take advantage of a pretrained conditional mean model to form a conditional generative model. Through empirical experiments, the paper shows superior performance of the proposed approach compared to state of the art uncertainty estimation techniques.

**Questions:**

- The classification extension generates y_0 as real numbers and then a combination of softmax and Brier score converts these generations to probabilities. What are the implications of this modeling assumption? I would also think approaches like discrete diffusion (Austin et al. 2021) can help systematically address the probability simplex concern raised here and I would love to hear authors' thoughts on this.
- Besides accuracy, classification experiments can benefit from reporting commonly used metrics in uncertainty quantification literature like ECE and NLL, which captures the whole distribution.
- The experiments are conducted using toy and small scale datasets and models. It would be good to see an extension of this approach to ImageNet style large scale settings.
- Referring to lines 141-144, the authors mention that PICP cannot capture if the learned quantiles systematically over/under estimate the true distribution. PICP does not have to utilize two sided quantiles. Isn’t this caveat mitigated if one studies the confidence intervals considering one sided quantiles?
- PICP is a common metric and it would be good to include it for UCI regression tasks. That will also help put the new proposed QICE metric in perspective.
- Algorithm 2 line 5. You might need to take the square root of \tilde{\beta_t}.
- Table 1 is not very informative and can easily go to an appendix. I would personally prefer to see more analysis on classification with possible extensions to other datasets. At least consider reporting the toy experiment results instead of this table. On a related note, toy experiment results are indicated to be recorded in Table 8 in Appendix A.3 but they are actually in Table 14.
- Line 239. Typo in “alignment”

Austin, J., Johnson, D.D., Ho, J., Tarlow, D. and van den Berg, R., 2021. Structured denoising diffusion models in discrete state-spaces. Advances in Neural Information Processing Systems, 34, pp.17981-17993.


**Limitations:**

The proposed approach makes assumptions on its extensions to classification settings and I believe the manuscript would benefit from a discussion on the implications of these assumptions. Furthermore, experimental results on classification could be improved as discussed in the above section.

**Strengths And Weaknesses:**

Originality: The work extends denoising diffusion probabilistic models for regression and classification tasks enabling uncertainty quantification. To my knowledge, this is the first study that attempts to take advantage of diffusion-based generative models for such tasks.

Quality: The paper is technically strong for regression but I have some questions about classification, which I list below. Also I think experimental results would be more compelling if the authors report commonly used metrics like NLL and ECE.

Clarity: The paper is well written and organized.

Significance: Uncertainty quantification is important for safe deployment of machine learning models and the paper advances state of the art by taking advantage of recently rising denoising diffusion models. I believe this is an important contribution to the field.

---

> ### Author Response · Authors · 2022-08-02
> **addressing questions and comments by Reviewer RrLR**
>
> Dear Reviewer RrLR,
>
> Thanks so much for your remarks and questions. We address them through our responses below.
>
> > Q1: Modeling assumption for classification and thoughts in discrete diffusion (Austin et al. 2021)
>
> A1: We appreciate you bringing this work to our attention. We have made connection to it in the added Related Work section. To construct our framework for classification, we assume the class labels in terms of one-hot vectors are from real continuous spaces instead of discrete ones. This assumption enables us to model the forward diffusion process and prior distribution at timestep T with Gaussian distributions, thus all derivations with analytical computation of KL terms, as well as corresponding algorithms, generalize naturally into the classification settings. The code for training and inference are exactly the same (in file diffusion_utils.py). Discrete diffusion models D3PMs fit conventional perception in classification tasks naturally by keeping the assumption of a categorical distribution. Therefore, the corresponding evaluation metrics like NLL can directly translate into such framework – we believe that by adopting the discrete space assumption, a better NLL metric can be achieved. Meanwhile, it would require a lot more changes to be made from our framework for regression tasks, including the choice of transition matrix, the incorporation of $x$ into the diffusion processes, as well as the addition of the auxiliary loss into the objective function – all of the above are classification-task-specific settings, and cannot be adopted with the existing framework for regression tasks.
> Besides the intention for consistency and generalizability across the two types of supervised learning tasks, we found that such construction gives reasonable results to access model prediction confidence at instance level – by directly use the prediction intervals obtained in the raw continuous space, i.e., before adopting the softmax function for conversion to probability space, we obtain the sharp contrast in PIW between correct and incorrect predictions, and can already achieve high accuracy by merely predicting the label with the narrowest PIW for each instance. After such conversion, the PIW contrast is reduced drastically, and the prediction accuracy by narrowest PIW is similar to a random guess.
>
> To recap, if achieving the best NLL and ECE for classification is the goal, then we think discrete diffusion models like Austin et al. (2021) could be excellent choices due to their use of the cross-entropy loss that is directed related to NLL and ECE; however, if the main goal is on modeling the confidence on predicted label probabilities, the proposed CARD works well, and it would be interesting to make a head-to-head comparison with discrete diffusion-based classification models that yet need to be developed.
>
> ------
>
> > Q2: ECE and NLL
>
> A2: We’ve updated the dataset for classification task to CIFAR-10, and reported NLL along with accuracy in comparison to the methods from Tomczak et al. (2021) in the revised paper. We’ve also obtained an ECE of 0.04&pm;0.00; however, since ECE wasn’t reported in Tomczak et al. (2021), we have yet to add it to the revised paper. Meanwhile, we would like to point out that ECE measures model calibration at subgroup level instead of instance level. We provide a detailed discussion in the metric of ECE in Appendix A.15. It is through such reasoning that motivated us to construct model confidence at instance level as we did in the paper.
>
> ------
>
> > Q3: Large scale datasets.
>
> A3: Please refer to our common response with ImageNet results.
>
> ------
>
> > Q4: Lines 141-144 for PICP.
>
> A4: We agree -- one-sided quantiles could mitigate this caveat, but then at least two one-sided quantiles might be needed.
>
> ------
>
> > Q5: PICP UCI regression tasks.
>
> A5: We’ve included the table reporting PICP for CARD along with all baseline methods in Appendix A.9. There is no clear best model in terms of this metric, as CARD along with PBP and MC Dropout all have $3$ wins. However, note that CARD's performance has been relatively stable across all methods.
>
> ------
>
> > Q6: Algorithm 2 line 5. Take the square root of $\tilde{\beta_t}$.
>
> A6: Thank you for pointing it out! We’ve made the modification in the revised paper. This is a typo from our end, and our code implementation is correct (the term is at the end of p_sample function in diffuse_utils.py file) – we’ve included the code as part of the Supplementary Material submission this time.
>
> ------
>
> > Q7: Move Table 1. to Appendix and more analysis on classification.
>
> Thank you for your suggestion -- we've moved Table 1 to Appendix A.8. We've obtained new results from CIFAR-10 and CIFAR-100 datasets, and plan to adjust the length in the regression toy experiment section so that we can provide more details in classification tasks.
>
> ------
>
> > Q8: Line 239. Typo in “alignment”
>
> A8: Thank you for capturing it – we’ve corrected the spelling in the revised paper.

---

> > ### Comment · Reviewer_RrLR · 2022-08-09
> > **Thank you authors**
> >
> > I thank the authors for their diligence in the rebuttal process. They have addressed my comments and I have increased my score to 6.

---

### Official Review · Reviewer_fUR1 · 2022-07-12

**Rating:** 5
**Confidence:** 3
**Soundness:** 2 fair
**Presentation:** 2 fair
**Contribution:** 2 fair

**Summary:**

This paper proposes diffusion-based approach on classification and regression tasks through conditional distribution modeling of y given x.
The authors also introduce a new metric for the regression task, called QICE based on PICP, which evaluates distribution similarity in terms of the percentiles.
The experiments are conducted on toy dataset, as well as UCI and Fashion MNIST, which show the effectiveness of proposed CARD compared to the baselines.

**Questions:**

- The major drawbacks of diffusion models are the model and time complexity. How different is the running time compared to the baselines? (both for the training and inference) Also, how different are the model complexities? The current experiment settings are relatively simple, hence, there might not be huge difference. However, when it comes to more complex situation, it can be a huge gap so that the performance is outweighed by its drawbacks.
- How does the behavior change as the denoising step increases, $t$ from $0$ to $T$? For example, in the case of Figure 1, and histogram or categorical probability change in the classification task.
- Could you compare CARD against neural processes? Also, I wonder about the quality of extrapolation in the regression task.

**Ethics Review Area:**

["I don’t know"]

**Limitations:**

The authors have discussed the limitations and potential negative societal impact in the appendix.

**Strengths And Weaknesses:**

- The paper discuss fundamental questions in the research community, which are classification and regression. Especially, the authors establish the conditional diffusion-based classification and regression.
- To overcome the drawback of PICP that it cannot represent the matching of the distribution centrality, the authors proposed a new metric, and The proposed QICE can be viewed as discretized version of NLL.
- The current version of experiments are relatively simple. While the advantage of diffusion models is that they can generate high-resolution images, it would be informative if the authors conduct experiments on such datasets.
- The description on the baselines are missing, CMV-MF-VI for example, in Table 5.
- The error bars are missing in the classification experiment, where the result seems that there is no huge difference among several models.
- The authors did not provide the code neither in the supplementary material nor as external link.

---

> ### Author Response · Authors · 2022-08-02
> **addressing questions and comments by Reviewer fUR1**
>
> Dear Reviewer fUR1,
>
> Thanks very much for providing your comments and questions. Please check below for our responses and answers.
>
> > Weakness 1: The current version of experiments are relatively simple.
>
> Response 1: We have updated experiment results on both small and large scale datasets including CIFAR-10/100, ImageNet-100/1K. Please check the results in our revision and in the common response above.
>
> ------
>
> > Weakness 2 & 3: The description on the baselines are missing; the error bars are missing in the classification experiment.
>
> Response 2 & 3: We’ve included the description in the baseline BNNs, and add the error bars for both accuracy and NLL in classification the CIFAR-10 dataset.
>
> ------
>
> > Weakness 4: The authors did not provide the code neither in the supplementary material nor as external link.
>
> Response 4: We’ve included the code and checkpoint in the Supplementary Material submission.
>
>
> ------
>
> > Q1: Model and time complexity.
>
> A1: We are currently re-running the baselines to measure the running time. We also reported the model size and computation complexity for various datasets in Appendix Section C. We would like to point out that we run these experiments with CARD on different servers with various GPU and CPU capacities (sometimes with shared tasks on the same device running at the same time), and different methods are implemented with different deep learning frameworks (CARD and GCDS in PyTorch; PBP, MC Dropout and Deep Ensembles in TensorFlow 2). Therefore, various factors could affect the actual running time.
>
> ------
>
> > Q2: Behavior change as the denoising step $t$ increases.
>
> We report the plots along with descriptions in Appendix A.13.
>
> ------
>
> > Q3:  Compare CARD against neural processes.
>
>
>
> A3: A short answer: CARD models $p(y|x,\mathcal{D}_i)$, while NP models $p(y|x,\mathcal{D}_o)$, where $\mathcal{D}_i$ and $\mathcal{D}_o$ represents in-distribution dataset and out-of-distribution dataset, respectively.
>
> To elaborate: although both classes of methods can be expressed as modeling $p(y|x,\mathcal{D})$, CARD assumes such $(x,y)$ comes from the same data-generating mechanism as the set $\mathcal{D}$, while NP assumes $(x,y)$ to be not from the same distribution as $\mathcal{D}$. While CARD fits in the traditional supervised learning setting for in-distribution generalization, NP is specifically suited for few-shot learning scenarios, where a good model would capture enough pattern from previously seen datasets so that it can generalize well with very limited samples from the new dataset.
>
> Furthermore, both classes of models are capable of generating stochastic output, where CARD aims to capture aleatoric uncertainty that’s intrinsic to the data (thus cannot be reduced), while NP can express epistemic uncertainty as it proposes more diverse functional forms at regions where data is sparse (and such uncertainty would reduce when more data is given). In terms of the conditioning of $\mathcal{D}$, the information of $\mathcal{D}$ is amortized into the network $\epsilon_{\theta}$ for CARD, while it is included as an explicit representation in the network that outputs the distribution parameters for $p(y|x)$. It’s also worth pointing out that CARD does not assume any parametric distributional form for  $p(y|x, \mathcal{D})$, while NP assumes a Gaussian distribution, and designs the objective function with such assumption.
>
> The concept and comparison between epistemic and aleatoric uncertainty is more thoroughly discussed in [What Uncertainties Do We Need in Bayesian Deep Learning for Computer Vision?](https://papers.nips.cc/paper/2017/file/2650d6089a6d640c5e85b2b88265dc2b-Paper.pdf) (2017) by Kendall and Gal, in which we quote, “Out-of-data examples, which can be identified with epistemic uncertainty, cannot be identified with aleatoric uncertainty alone.” We acknowledge that modeling OOD uncertainty is an important topic for regression tasks; however, we design our model to focus on modeling aleatoric uncertainty in this paper. We appreciate your suggestion in looking at this direction, and we plan to explore CARD’s ability in extrapolation as part of our future work.

---

> > ### Comment · Reviewer_fUR1 · 2022-08-08
> > **Thanks**
> >
> > Dear authors,
> >
> > Thank you for your clarification. After reading other reviewer's comments, I decide to keep my score.

---

### Author Response · Authors · 2022-08-03
**Common Response to All Reviewers**

Dear reviewers,

We thank you all for taking your time to review our submission, and appreciate your valuable insights and comments. Here we provide some common responses:

> The common suggestion on using large scale datasets for classification tasks.

Response: We’ve obtained experimental results on both CIFAR-10 and CIFAR-100 datasets with similar conclusions presented in our revised paper (i.e., CARD improves accuracy from the pre-trained base classifier $f_{\phi}(x)$; class with narrowest prediction interval can already provide a great accuracy; prediction confidence by the model at instance level through: sharp contrast at both global and class level in prediction interval width of the true class for correct vs. incorrect predictions, sharp contrast at both global and class level in accuracy for rejected vs. not-rejected instances by $t$-test). We have moved the results on FashionMNIST to Appendix A.3 and used the saved space to add the results on CIFAR-10 – these newly added results provide further support to the conclusions of the original submission. Due to time constraints and limitations in computational resources, we are currently still running the model on the ImageNet dataset, and will report the results as soon as we obtain them.

> Revision in the paper and appendix.

We currently mark the text with main revision and addition (e.g., replacing FashionMNIST dataset with CIFAR-10 in the paper, additional metric reports in the Appendix, etc.) in blue.

> A few things to add in the upcoming week.

We plan to continue updating the paper in a few places:

* add more plots and ablation study, as well as more detailed analysis in classification tasks, suggested or implied by different reviewers (we already have the results but it takes some time to summarize them into the appendix)
* upload checkpoints of trained models

---

### Author Response · Authors · 2022-08-05
**Updates on Revised Paper and Appendix**

Dear reviewers,

We’ve just uploaded an updated paper and appendix revision, including the following notable changes and additions:

* a more detailed analysis for the classification task in Section 3.2 (utilizing the extra space after moving the toy regression experiment analysis into Appendix A.12)
* an updated related work section, currently placed at Appendix A.2
* ablation study on CIFAR-10 dataset, with different prior distribution settings, in Appendix A.11
* the behavior change in samples as t increases, in Appendix A.13
* the metric change during test time as t decreases, in Appendix A.14

Note: We've re-arranged our submission so the revised paper now contains the appendix, and Supplementary Material now is the zip file of our code.

---

### Author Response · Authors · 2022-08-09
**Updated Response to All Reviewers (Part. 1)**

Dear reviewers,

We appreciate your valuable feedbacks. We are glad to let you know that in addition to the CIFAR 10 results reported in our previous revision, we have now obtained some initial results on evaluating CARD on CIFAR-100, ImageNet-100, and ImageNet-1K, which all consistently show that CARD not only outperforms its corresponding deterministic classifier trained on the same dataset in terms of classification accuracy, but also provides uncertainty estimation that is missing from using a deterministic classifier. In addition, we have performed additional experiments to study the effect of conditioning on $x$ in the forward diffusion process. We have updated the manuscript accordingly to include these new results, which are currently placed towards the end of Appendix A for easier reference. In what follows, we summarize these new results for your convenience:


### Classification on large-scale datasets (CIFAR-10, CIFAR-100, ImageNet-100, ImageNet-1K)

We first report the classification accuracy:

For CIFAR-10 and CIFAR-100, we include comparison to all baselines applied in our paper (reported in Tomczak et al. 2021) in Table 4 under Sec. 3.2.2, and in Table 20 under Appendix A.16, respectively.

We further evaluate the classification accuracy of CARD on ImageNet-100 and ImageNet-1K, whose results are also presented in Appendix A.17 and A.18. We can see CARD still outperforms the pre-trained classifier $f_{\phi}$ under a moderately increased parameter size.

> Table R2: Comparison of top-1 classification accuracy on ImageNet-100 and ImageNet-1K
|                | Parameter | ImageNet-100 | ImageNet-1K |
|----------------|-----------|--------------|-------------|
| Deterministic  | 25.6M     | 79.40        | 73.87       |
| CARD           | 32.7M     | 82.34        | 74.32       |

|                             | Parameter | ImageNet-100 | ImageNet-1K|
|-----------------------------|-----------|--------------|-------------|
| Deterministic               | 25.6M     | 79.40        | 73.87       |
| CARD                        | 32.7M     | 82.34        | 74.32       |

Besides the classification accuracy, we provide the model uncertainty measure in Table R3 (similar to Table 5 in our paper), which also includes the accuracy of predicting by the class label with the narrowest PIW:

> Table R3: PIW (multiplied by 100) and $t$-test results for the CIFAR-10, CIFAR-100, ImageNet-100, and ImageNet classification tasks.

| Data         | Class      | Accuracy  | PIW (Correct) | PIW (InCorrect) | Accuracy by PIW | Accuracy (Rejected) | Accuracy (Not-Rejected) | Count (Not-Rejected) |
|--------------|------------|-----------|---------------|-----------------|---------------|-----------|---------------|-----------------|
| CIFAR-10     | overall    | 91.79    | 0.67        |  3.20           | 87.84 | 91.25   | 42.86          |  63      |
|              | most acc.  | 98.50    | 0.39     | 2.08            |            |            |           |               |             |
|              | least acc. | 74.80    | 1.37          | 3.26        |            |            |           |               |             |
|              |            |           |               |                 |            |            |           |               |
| CIFAR-100    | overall    | 71.42 | 0.59        | 3.91            | 60.53 | 71.56 | 35.90        | 39            |
|              | most acc.  | 95.00 | 0.16        | 1.92          |            |            |           |             |                 |
|              | least acc. | 44.00 | 5.09        | 5.84            |            |            |           |            |                |
|              |            |           |               |                 |            |            |           |               |
| ImageNet-100 | overall    | 82.34 | 2.06        | 13.73        | 68.64 | 82.90 | 34.48        | 58           |
|              | most acc.  | 98.00 | 0.72        | 8.06            |            |            |           |           |                 |
|              | least acc. | 42.00 | 6.79        | 14.15           |            |            |           |               |                 |
|              |            |           |               |                 |            |           |               |
| ImageNet-1K  | overall    | 74.28     |0.65           | 3.11                | 69.22   | 74.63     | 24.93           | 349                |
|              | most acc.  | 98.00     | 0.27           |     2.80            |            |            |           |               |                 |
|              | least acc. | 8.00     | 20.10           |      50.07          |            |            |           |               |

---

### Author Response · Authors · 2022-08-09
**Updated Response to All Reviewers (Part. 2)**

Due to the number of classes besides the CIFAR-10 dataset, we do not report the metrics for every class. Instead, for PIW, we report the PIW of true label for correct and incorrect predictions, for the entire test set as well as only the most and the least accurate class; for $t$-test results, we only report the accuracy for instances with rejected $t$-test and those with not-rejected $t$-test, as well as the total count of not-rejected cases, for the entire test set.




### The effect of conditioning on $x$ in the forward diffusion process


When we add the condition of $x$ to the forward process, we have $q(y_t|y_{t-1},x)$ instead of $q(y_t|y_{t-1})$, which results in a prior distribution of $\mathcal{N}(f_\phi(x), I)$ by our formulation in Eqn. (8) in paper. Otherwise, we have a prior of $\mathcal{N}(\mathbf{0}, I)$, which can also be interpreted as having $f_\phi(x) = 0$, *i.e.,* the forward process is independent of $x$, thus $q(y_t|y_{t-1}, x) =q(y_t|y_{t-1})$. We conduct ablation studies on the effects of these two prior distribution settings. Please refer to Appendix A.10 and Table 16 for more details.

Moreover, we observe that a $\mathcal{N}(f_{\phi}(x),I)$ setting makes the model improve faster to a satisfactory performance level. Please refer to Appendix A.11 and Figure. 3.

We also provide an ablation study on different parameterizations of the diffusion noise $\epsilon_{\theta}$ network.  Detailed model configurations and model parameter measures are also included. Please refer to Table 17 in Appendix A.11 and Table 22 in Appendix C for more details.

### The behavior of samples and model performance through diffusion process

We plot the evolution in samples of both forward and reverse diffusion processes during training on toy regression dataset (linear regression, full circle), as well as those during test time on UCI Boston dataset. Please refer to Appendix A.13 and Figure. 4-7 for a detailed analysis.

We also plot the development of model performance in terms of evaluation metrics through the reverse diffusion process on UCI Boston dataset. Please refer to Appendix A.14 and Figure. 8 for a detailed analysis.

### More analysis on classification in the paper

Besides the extensions of the classification algorithm to multiple large scale datasets, we also provide a more detailed analysis on the classification task in the paper, after moving the majority of the descriptions in toy regression tasks to Appendix A.13. Please refer to Sec. 3.2 and Appendix A.15 for the corresponding revision.

### Checkpoint link

We include the checkpoints for UCI regression and all classification tasks [here
](https://drive.google.com/drive/u/8/folders/1hWT3kW7KssDnfBMJVMedFC14JfELmczM).

---

### Meta-Review · Area_Chair_bot6 · 2022-09-02

**Recommendation:** Accept
**Confidence:** Less certain

**Metareview:**

This paper proposes a different approach to uncertainty quantification in regression and classification problems. It extends denoising diffusion probabilistic models in combination with a pre-trained conditional mean model to provide a conditional generative model. The proposed approach is evaluated on UCI regression tasks and CIFAR10 classification. Additional results on CIFAR100 and ImageNet were presented during the rebuttal period and additional analyses.

The paper received some (relatively weak) support among reviewers. All the reviewers agree the paper proposes an interesting and novel formulation of uncertainty-quantified regression and classification models. The main issues raised by the reviewers were that of (i) conducting experiments on larger datasets, (ii) more analysis in the classification case, and (iii) the effect of conditioning on x in the diffusion processes. I believe the authors have provided additional significant empirical evidence of the benefits of their approach and some of the reviewers updated their scores accordingly. Given this, the potential impact of the paper, the novelty nature of the contribution, and despite the seemingly weak scores, I believe there are no significant concerns remaining about this paper and it is worth presenting in its current form to the NeurIPS community.


**Award:**

No

---

### Decision · Program_Chairs · 2022-09-14

Accept